# Assessing and enhancing robustness of active learning strategies to spurious bias

## Abstract

We focus on investigating the performance of common active learning (AL) algorithms under spurious bias and designing an AL algorithm that is robust to spurious bias. Spurious bias refers to the bias created when certain potentially simpler, task-irrelevant attributes in the training set are highly correlated with the target labels. Spurious bias can occur if the sample we use for analysis is not representative of the population, some samples are overrepresented while others are underrepresented. The AL criteria share similarities with approaches to addressing spurious correlations in passive settings. Hence, with an appropriately defined acquisition function, a sample-efficient framework can be established to effectively handle spurious correlations. Inspired by recent works on simplicity bias, we propose **D**omain-**I**nvariant **A**ctive **L**earning (DIAL) which leverages the disparity in training dynamics between overrepresented and underrepresented samples, selecting samples that exhibit "slow" training dynamics. DIAL involves no excessively resource-intensive computations as it only relies on training checkpoints to estimate the dynamics of the samples, making it more scalable for addressing real-world spurious correlation problems with AL.

## 1 Introduction

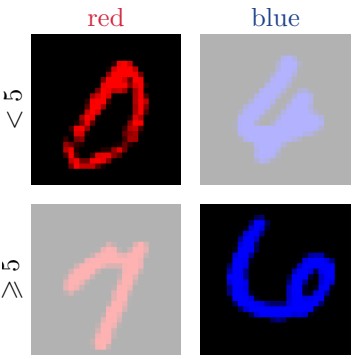

Figure 1: Example of spurious correlation in Coloured MNIST. Digits less than 5 are mostly appeared in blue, while digits greater than 5 are mostly appeared in red.

Spurious correlations refer to the scenario where certain (potentially simpler) task-irrelevant attributes in the training set are highly correlated with the target labels. Consider the scenario where a deep neural network (DNN) is trained to distinguish between if the coloured digit in the image is less than 5 or greater than 5, as shown in Figure 1. In the training dataset, an unintended sampling bias might emerge, leading to a situation where the majority of digits less than 5 are predominantly red, while the majority of digits greater than 5 tend to be blue. This sampling bias inadvertently introduces a spurious correlation between the digits and the colour attribute. As a result, the trained DNN may mistakenly learn to associate the presence of a certain colour with a particular object class, leading to erroneous predictions when faced with images featuring digits of different colours. For example, during deployment, the model might classify any blue digit as $\geqslant 5$ and any red object as $< 5$, regardless of other features, such as shapes and edges. This issue can have adverse effects in real-world applications – relying on these false associations can result in flawed predictions, inaccurate analyses, and misguided actions, particularly in critical domains such as healthcare (Oakden-Rayner et al., 2020) and social sciences (Angwin et al., 2016).

One common category of existing approaches for learning robust DNNs (Sagawa et al., 2020; Liu et al., 2021; Sohoni et al., 2020; Nam et al., 2020) rely on identification of underrepresented samples, often achieved through explicit labels or estimation (more detailed discussions are provided in Section 2). This process resembles the acquisition step in the active learning (AL) loop where the informativeness is based on the representativeness of samples in the labelled pool. The key distinctions between standard AL methods and

existing approaches to spurious correlation are as follows: (1) the former assesses informativeness solely based on sample features, while the latter evaluates not only based on features but also considers label information (in our illustrative scenario, class label – cars and bicycles – and/or attribute label – red and blue); (2) after evaluation, the former appends newly labelled samples to existing labelled dataset while the latter assigns importance weightings for subsets of samples. Those distinctions, though, are blurry. The phase when passive approaches to spurious correlation identify underrepresented samples essentially corresponds to a single-step active acquisition but without explicit labels. Furthermore, the act of appending samples to an existing dataset is equivalent to increasing the importance of specific populations in the dataset (resampling versus reweighing).

Tamkin et al. (2022) demonstrated that uncertainty-based AL methods can inherently address various subpopulation shift problems. Their research showed that AL methods improve both overall performance and the performance of underrepresented subgroups compared to random sampling. This improvement arises from the increased of labelled disambiguating samples – those whose spurious attributes do not align with their labels in the labelled pool – from the unlabelled pool by AL methods. However, uncertainty-based acquisition can sometimes miss these disambiguating samples if their predictions are overly confident due to reliance on spurious features, resulting in low uncertainty and causing these samples to be neglected in the AL loop. Consequently, the robustness improvement is limited. Therefore, this work aims to design a deep learning AL method that can accurately identify underrepresented/disambiguating samples.

Our motivation stems from recent research on *simplicity bias*, where DNNs trained with stochastic gradient descent (SGD) tend to prioritize learning simple features over complex ones (Shah et al., 2020; Nakkiran et al., 2019). Consequently, the DNN will exhibit invariance to complex features, leading to potential detrimental effects on generalization, especially when the simple features are completely irrelevant to the task. The lack of robustness under spurious correlation arises from the bias towards much simpler (and thus, easier-to-learn) features over task-intrinsic ones (Teney et al., 2022; Vasudeva et al., 2023; Bell & Sagun, 2023). In essence, such classifiers adopt simplistic decision rules heavily influenced by the statistical bias present in the training samples, e.g., relying on the colour instead of other features. Several works in passive settings have leveraged this phenomenon to design methods that demonstrate strong effectiveness (Liu et al., 2021; Murali et al., 2023; Yang et al., 2024). Moreover, from the perspective of training loss profiles, Nam et al. (2020) showed a discrepancy in terms of training dynamics, where the loss of samples aligned with the bias converges faster than those unaligned.

Inspired by these observations, we introduce **D**omain-**I**nvariant **A**ctive **L**earning (DIAL), which utilizes training dynamics as a proxy to assess the informativeness of unlabelled samples. Our experiments demonstrate that DIAL effectively selects underrepresented samples from the labelled pool, thereby enhancing the robustness of the classifier. Furthermore, although the concept of training dynamics is often associated with gradients, DIAL does not require gradient operations but instead relies solely on feedforward inference during sample acquisition, making it efficient in the context of deep learning.

Our contributions in this work can be summarized as follows:

- We show the potential of addressing spurious correlations in the sample-efficient manner by means of AL.
- Drawing inspiration from simplicity bias, we proposed DIAL, a deep AL method designed to mitigate spurious bias. DIAL is easy to implement and highly compatible in deep learning.
- We perform extensive experiments on real-world datasets with spurious correlations, demonstrating that our proposed method, DIAL, achieves better robustness than existing AL methods.
- We perform ablation studies to investigate the effectiveness of DIAL across different hyperparameter settings and AL configurations.

## 2 Related work

**Active learning.** The primary goal of AL algorithms is to query labels for the most informative samples. These criteria for informativeness are broadly categorized into two groups (Settles, 2012): *uncertainty-based* and *representation-based* sampling. One classic notion of uncertainty-based sampling involves assessing the

distance of the sample to the decision boundary, including the margin (Roth & Small, 2006), the entropy of softmax outputs (Wang & Shang, 2014), and the maximum class predictive score (Wang & Shang, 2014). In the context of probabilistic formulation, another notion is derived from the information-theoretic perspective where Roy & McCallum (2001) uses the expected information gain, and Houlsby et al. (2011) measures the mutual information between the model posterior and the samples. Gal et al. (2017) incorporates Monte Carlo dropout (Gal & Ghahramani, 2016) to overcome the expensive computation of mutual information of (Houlsby et al., 2011) for DNNs and Kirsch et al. (2019) further extends the approach to consider the dependencies between samples in the batch. Representation-based strategies, on the other hand, adopt the notion of representativeness as the measure for informativeness. Settles et al. (2007) evaluates representativeness in the gradient space which selects samples with maximum norm of gradient in order to maximize the change of model's parameter. Sener & Savarese (2018) and Geifman & El-Yaniv (2017) label samples that exhibit diversity w.r.t. the current labelled pool with the goal of constructing a proxy for the complete dataset. Similarly, Ash et al. (2020) seeks to promote diversity using gradient embeddings. A closely related study to our proposed method is presented in Jung et al. (2022), where they proposed utilizing snapshot-ensemble (Huang et al., 2016) instead of commonly used methods such as deep ensemble (Lakshminarayanan et al., 2017) during the acquisition phase. In Jung et al. (2022), a cosine annealing learning rate scheduler (Loshchilov & Hutter, 2017) with large initial learning rate is used to enforce ensure that each snapshot converges to diverse local minima. In contrast, our proposed method does not use the cyclical learning rate scheduler, as it leverages the training dynamics of DNN to assess the informativeness of unlabelled samples.

**Active learning for fairness.** Conventional AL aims to enhance model performance by selectively querying the most informative samples from the unlabelled pool. However, this often overlooks the fairness aspect, potentially perpetuating or even exacerbating biases present in the data. To address this, several works have proposed fairness-aware AL methods. Similarly, Anahideh et al. (2022); Sharaf et al. (2022); Fajri et al. (2024) incorporate fairness constraints directly into the sample selection process, aiming to achieve trade-off between fairness and model performance. However, these approaches necessitate explicit access to bias attributes in the labelled or even unlabelled pool. This requirement can limit their practicality in real-world scenarios where access to bias attributes might be restricted due to privacy regulations. In this work, we aim to eliminate this requirement, aligning with the current research trends in passive settings.

**Subgroup robustness.** Numerous methods are available to tackle bias caused by spurious correlations in the dataset. These methods share a common objective: to mitigate bias by improving the *worst-group performance* so that the disparities between overall performances is minimized. Examples include subgroup robust approaches (Sagawa et al., 2020; Nam et al., 2020; Liu et al., 2021; Yao et al., 2022; Sohoni et al., 2020) and representation learning approaches (Arjovsky et al., 2020; Zemel et al., 2013). Some of these methods require explicit spurious attribute information (e.g., (Sagawa et al., 2020; Yao et al., 2022)), thereby limiting their use cases. In contrast, our proposed algorithm requires no additional information beyond the class label. Specifically, upon the AL algorithm's selection of samples, the expert is only required to annotate them with class labels.

## 3 Background

**Notation.** We consider classification problems. Given a set of samples $\mathcal{D} = \{(x_1, y_1), \ldots, (x_n, y_n) \mid x \in \mathcal{X}, y \in \mathcal{Y}\}$, where $\mathcal{X}$ denotes the feature space and $\mathcal{Y}$ denotes the labels, and a loss function $\ell : \mathcal{Y} \times \mathcal{Y} \to \mathbb{R}$, the task is to learn a DNN $f_\theta : \mathcal{X} \to \mathcal{Y}$ parameterized by $\theta$ via empirical risk minimization (ERM):

$$\theta^* = \arg\min_{\theta \in \Theta} \sum_{i=1}^n \ell\left(f_\theta(x_i), y_i\right) \tag{1}$$

We omit the symbol $\theta$ when the context is clear. Furthermore, we express the DNN obtained at time $t$ as $f_{\theta_t}$.

**Active learning.** In pool-based AL scenario, starting with a labelled pool $\mathcal{D}_L$ the algorithm sequentially queries an oracle for annotations of some unlabelled samples from $\mathcal{D}_U = \{x_1, \ldots, x_m\}$. We overload the notation by using $f_{\mathcal{D}_L}$ to denote the DNN obtained on the labelled pool $\mathcal{D}_L$. Formally, at each AL iteration, the

active learner chooses some samples from the unlabelled pool $\mathcal{Q} \in \mathcal{D}_U$ (typically conditioned on $f_{\mathcal{D}_L}$). Following this selection, the oracle $\mathcal{H}$ (i.e., human annotator) assigns labels for $\mathcal{Q}$: $\widetilde{\mathcal{Q}} := \mathcal{H}(\mathcal{Q}) = \{(x, h(x)) \mid x \in \mathcal{Q}\}$, where $h(x) = y$ is the ground truth label of sample $x$. Subsequently, the pool gets an update: $\mathcal{D}_L \leftarrow \mathcal{D}_L \cup \widetilde{\mathcal{Q}}$ and $\mathcal{D}_U \leftarrow \mathcal{D}_U \setminus \mathcal{Q}$. This process continues until a stopping criterion is met, e.g., when a labelling budget is exhausted.

**Spurious bias.** Typically, one would anticipate learning a classifier that makes predictions based on semantic features (i.e., features that are relevant to the prediction task). However, in many instances, the feature domain comprises other meaningful attributes that are irrelevant to the task such as image background (Sagawa et al., 2020), and demographic identities (Liu et al., 2015; Borkan et al., 2019). We refer to these attributes as *spurious attributes*, denoted by $\mathcal{S}$, and we define subgroups as $g \in \mathcal{G} := \mathcal{Y} \times \mathcal{S}$. Formally, spurious correlation refers to the scenario where a statistical dependency between the task and the spurious attributes is observed solely within the training set[1] (Yang et al., 2023):

$$\mathbb{P}_{\mathrm{tr}}(X, Y, S) \propto \mathbb{P}_{\mathrm{tr}}(Y \mid S) \, \mathbb{P}_{\mathrm{tr}}(S). \tag{2}$$

This correlation, however, is absent in the test set:

$$\mathbb{P}_{\mathrm{te}}(X, Y, S) \propto \mathbb{P}_{\mathrm{te}}(Y) \, \mathbb{P}_{\mathrm{te}}(S). \tag{3}$$

More specifically, in the training set, we have $\mathbb{P}_{\mathrm{tr}}(Y \mid S) \neq \mathbb{P}_{\mathrm{tr}}(Y)$. Learning becomes challenging in the presence of spurious correlation because the classifier may shift its reliance from semantic features to spurious attributes, leading to errors during deployment.

To assess robustness with respect to spurious bias, we adopt two key performance metrics in our analysis. The first metric is the *average accuracy*: $\mathbb{E}_{(x,y)} [\mathbb{1}[f(x) = y]]$, which provides an assessment of the overall model performance. The second metric is the *worst-group accuracy* (Sagawa et al., 2020) defined as the "accuracy" of the worst-performing subgroup: $\min_{g' \in \mathcal{G}} \mathbb{E}_{(x,y)|g=g'} [\mathbb{1}[f(x) = y]]$. More formally, the worst-group accuracy is known as the *worst true positive rate of one group versus the other groups*. This is a commonly used evaluation measure in the field of subgroup robustness (Idrissi et al., 2022; Yang et al., 2023). In line with Nam et al. (2020), we term samples correctly classified by the bias attribute based on the biased training set as *bias-aligned* (BA) samples, and those misclassified as *bias-conflicting* (BC) samples. Throughout the paper, for more broader context, we use the terms *underrepresented* and *overrepresented* to refer to BC and BA samples, respectively.

**AL for spurious bias.** The primary reason for poor robustness under spurious correlations is due to the underrepresentation of certain subgroups in the training set. Consequently, many passive approaches address this issue directly by augmenting the representation of underrepresented samples through upsampling. In this study, we adopt a similar approach in the active way, i.e., improving the representation of underrepresented samples through the acquisition of new samples. In passive frameworks, the criteria for upsampling typically rely on some proxy measure, commonly the predictive error (Sagawa et al., 2020; Liu et al., 2021; Duchi & Namkoong, 2021). However, this approach is unfeasible in AL, as neither the target labels nor the bias labels are observable before acquisitions. This raises the question: how can we identify underrepresented samples from the unlabelled pool without any labeling information? The following section describes our motivation and principle behind our proposed method.

## 4 Limitation of uncertainty-based active learning in enhancing robustness

The goal of uncertainty-based AL methods is to minimize model uncertainty, thus enhancing overall performance (Settles, 2012; Huang et al., 2014). This is achieved by selecting samples that the current model finds less certain about. While there are various metrics to quantify uncertainty, they all share a common principle: samples close to the decision boundary tend to be more informative.

---

[1] We assume that the covariate distribution is invariant between the training and test environments, i.e., $\mathbb{P}_{\mathrm{tr}}(X \mid Y, S) = \mathbb{P}_{\mathrm{te}}(X \mid Y, S)$.

Figure 2a illustrates a 2D binary classification task where the features are generated by Gaussian distributions $x \sim \mathcal{N}\left(\left[\mu_1, \mu_2\right], \Sigma\right)$, with identical diagonal covariance matrix $\Sigma = \sigma^2 I$ for every subgroup. The class label $Y$ is determined by the 1st feature coordinate $x_1$ (x-axis), and the spurious attribute $S$ is defined by the 2nd feature coordinate $x_2$ (y-axis). The decision boundary for the optimal and robust classifier will be the vertical line at the center. All subgroups are linearly separable along the central axis. Due to spurious correlations, samples with $y = \square$ are more frequently observed in red while samples with $y = \circ$ are more frequently observed in blue, making other combinations rare. Hence, the BA subgroups are $\{(y = \square, s = \text{red}), (y = \circ, s = \text{blue})\}$, while the BC subgroups are the complementary pairs $\{(y = \square, s = \text{blue}), (y = \circ, s = \text{red})\}$. For more intuitive explanation, in this section, we use *minority* to refer to the BC subgroups and *majority* to refer to the BA subgroups. We then train a linear classifier for the task. To emulate the simplicity bias phenomenon during training, we incorporate simplicity regularization.[2]

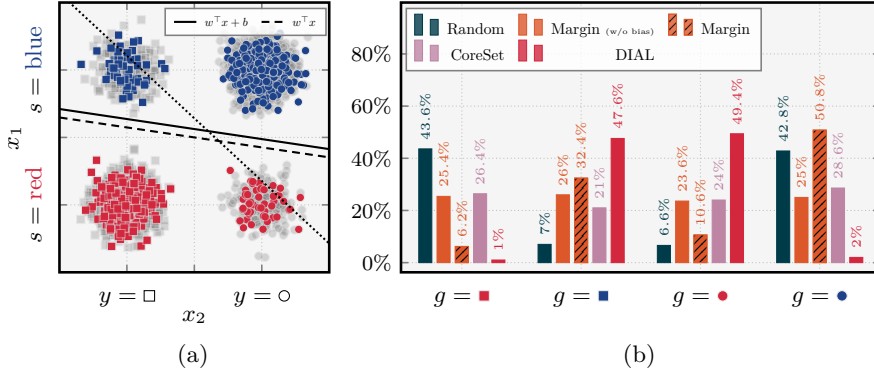

(a)           (b)

Figure 2: (a) 2D binary toy example. The line ⋯⋯ represent the decision boundary of the classifier after acquiring all minority samples[3]. The majority (resp., majority) is on the off-diagonal (resp., diagonal). (b) The distribution (over subgroups) of the queried batch.

One potential limitation of uncertainty-based AL methods is their reliance on predictive uncertainty as the criterion (Roth & Small, 2006; Wang & Shang, 2014), which can be biased if the predictive model itself is biased. For instance, in Figure 2a, the uncertainty measure primarily reflects the classifier's uncertainty about the spurious attribute (colour) rather than the target (shape). Consequently, a minority sample might exhibit low uncertainty even if it is misclassified. In this illustrative example, we consider two linear classifiers: one with additive bias and one without.

When employing the unbiased classifier, the mean of the minority samples in the unlabelled pool is closer to the decision boundary compared to the majority samples. However, due to their disproportion, there are significantly more majority samples with similar levels of ambiguity as the minority samples. Hence, the AL agent queries a small number of minority samples alongside many majority samples.

Conversely, the biased classifier shifts the decision boundary towards certain subgroups. As shown in Figure 2b, uncertainty sampling enhances the representation of minority compared to random sampling. However, this behaviour is substantially influenced by the classifier's characteristics. For example, a biased classifier makes the blue subgroups more uncertain than the red ones, resulting in uncertainty sampling favouring these subgroups and leading to the acquisition of a higher proportion of majority samples (blue rectangles) and a smaller proportion of minority samples (red circles). We observed similar behaviour on the real dataset (see Section 5.3). On the other hand, our proposed method, DIAL, is able to acquire a significantly higher proportion of minority samples, while ensuring a balanced representation of the minority subgroups. As a result, spurious correlations are mitigated, and the robustness across subgroups is improved, aligning closer to the ideal scenario depicted in Figure 2a (decision boundary of ⋯⋯ ).

**Why is it important to prioritise minority samples?** The lack of subgroup robustness is due to the bias introduced by the underrepresentation of certain subgroups. The intuitive approach to reduce this bias is to balance the subgroups. This requires acquiring relatively more minority samples than majority ones to "fill the gaps". The ideal "filler" is a dataset with a complementary distribution to the existing labelled set (i.e.,

---

[2]We introduce a regularization term $|w_1|$ to the loss function, simulating that learning $x_1$ is more challenging than $x_2$. Note that this simplicity bias is the key cause of poor robustness under spurious correlations, as evidenced by recent studies. It can be shown that enforcing a complexity bias through a $|w_1|$ term would yield an unbiased classifier, even in the presence of data bias.

[3]This is a hypothetical scenario to illustrate the potential of AL in enhancing robustness by acquiring minority samples.

large amount of minority samples and small amount of majority samples). While diversity-based strategies, such as CoreSet (Sener & Savarese, 2018), aim to acquire diverse samples and often achieve balanced subgroup representation (as shown in Figure 2b), their ability to improve subgroups robustness can be limited. This is because they also acquire equal amounts of majority samples, diluting the focus on the minority. Therefore, diversity alone is insufficient for our problem; the acquisition method must specifically favour minority to effectively enhance robustness.

Despite this limitation, uncertainty-based approaches still offer advantages over random sampling. However, they might exhibit slow progress in improving robustness due to sampling bias. To address this challenge, we require an approach capable of identifying and isolating minority samples. By prioritising these samples, we can expedite the development of a more robust classifier through AL.

## 5 Motivation

### 5.1 Bias-aligned samples are learned faster

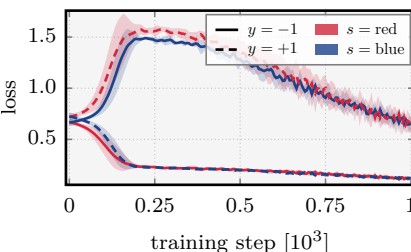

Figure 3: Training loss of aggregated over different subgroups. The colour represents the bias (digit colour).

When training a DNN with the ERM objective on a biased dataset, a notable phenomenon arises: the loss of the BA groups decreases much more rapidly than that of the BC groups, as discussed in Nam et al. (2020); Qiu et al. (2023). This discrepancy arises because the model tends to learn a decision rule that heavily relies on the bias in the early training stage. As a result, the loss for BA subgroups rapidly approaches zero, while the loss for BC subgroups remains high. To demonstrate this phenomenon, we consider a binary classification task involving a binary spurious attribute. We construct a toy example using a binarized MNIST dataset, where digits less than five are labelled as the negative class, while those above are labelled as the positive class. To simulate spurious bias, we assign colours ($\mathcal{S} = \{\text{red}, \text{blue}\}$) to digits

(e.g., $\{\text{🄋🄵🄸🄸}\}_{s=\text{red}}^{y=-1}$, $\{\text{🄸🄵🄸🄸}\}_{s=\text{blue}}^{y=-1}$, $\{\text{🄸🄵🄸🄸}\}_{s=\text{red}}^{y=+1}$, $\{\text{🄸🄵🄸🄸}\}_{s=\text{blue}}^{y=+1}$), with the training set primarily featuring *negative-class* digits in red and *positive-class* digits in blue which establishes a statistical relationship between colour and target label. As depicted in Figure 3, the loss of BA subgroups shows notably quicker convergence compared to the BC subgroups. This suggests that BC or underrepresented samples can be readily identified by their loss with a model from early training stages (Liu et al., 2021). However, this methodology becomes impractical in AL scenarios where labels are unavailable. Moreover, pseudo-labels, often utilized as substitutes for ground truth labels in AL algorithms (Ash et al., 2020; Wang et al., 2022), may not be suitable, as the model could confidently make incorrect predictions based on incorrect contexts, making the samples distinguishable by loss (further discussion is provided in Section 5.2). Nevertheless, this discrepancy in learning dynamics serves as a useful indicator to discern sample membership, which is the primary motivation behind our approach. This is explored in the next section.

### 5.2 Inference of learning dynamics

The training dynamics can be further interpreted from an inference perspective. Specifically, predicting an unseen BA sample is anticipated to exhibit sustained low variance throughout the entire training process, while the prediction of a BC sample gradually converges towards the ground truth over time.

Figure 4 presents the view of the loss landscape alongside the predicted probability of unseen samples from multiple subgroups in the parameter space. Distinct behaviors emerge for BC samples ($\{(y = -1, \text{blue}), (y = +1, \text{red})\}$) compared to BA samples ($\{(y = -1, \text{red}), (y = +1, \text{blue})\}$) as the model converges to the local minimum. In the case of BA samples, the model consistently delivers correct predictions with minor variations in confidence. Conversely, for BC samples, the model initially tends to make incorrect predictions, and significant fluctuations in predictions occur until it eventually converges to the ground truth. This exploration underscores the intricate dynamics involved in handling underrepresented samples offering an insight into designing an acquisition function based on the learning dynamics. Furthermore, to accurately capture the dynamics, one approach is to measure the rate of change of the output of the model w.r.t. training

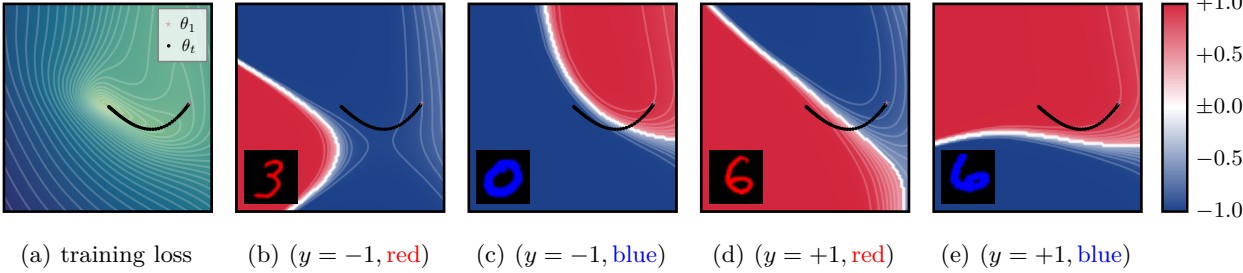

(a) training loss     (b) $(y = -1, \text{red})$     (c) $(y = -1, \text{blue})$     (d) $(y = +1, \text{red})$     (e) $(y = +1, \text{blue})$

Figure 4: Visualization of multiple metrics in the parameter space. (a) illustrates the training loss surface. The predicted probability of test samples (displayed in the corners) from different subgroup is depicted in (b) and (e) for BA subgroups, and (c) and (d) for BC subgroups, where the sign indicates the predicted label (+ for positive and − for negative classes). The learning trajectory is represented by ● with ⋆ denoting the initial point. The model traverses through a significantly greater number of levels in the probability map for BC subgroups compared to BA subgroups, indicating more pronounced predictive variations in the learning trajectory for BC samples.

time, denoted as $\partial f(x)/\partial t$ for a given input $x$. This quantity can be computed analytically using the neural tangent kernel (NTK) framework (Lee et al., 2019; Jacot et al., 2018). Additionally, Wang et al. (2022) also incorporated the notion of learning dynamics into their acquisition explicitly computed using NTKs. However, this analytical approach faces practical challenges, particularly in the context of large-scale DNNs with numerous parameters. The computational complexity escalates significantly, particularly in AL scenarios where the computation must scale with the number of AL loops. This computational burden highlights the necessity of exploring alternative methodologies. Thus, in this paper we aimed to propose an alternative measure that offers scalability and computational efficiency.

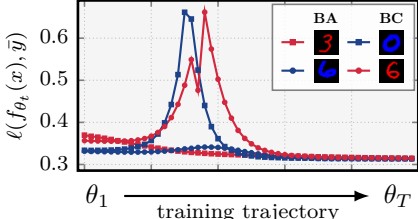

Figure 5: Pseudo-loss progression along the training trajectory from initial (left) to end (right).

**Pseudo-labels can be misleading.** Moreover, we explore the feasibility of leveraging pseudo-loss for identification of underrepresented samples. Figure 5 visualises the pseudo-loss $\ell(f_\theta(x), \bar{y})$ across different training steps, where $\bar{y} = \arg\max f_\theta(x)$ denotes the pseudo-label assigned to individual sample $x$. In the initial training phase, the differences are primarily observable among the different spurious attributes $\mathcal{S} = \{\text{red}, \text{blue}\}$. A clearer differentiation emerges between BA and BC subgroups, ultimately reaching disparate levels, as training progresses. Eventually, with more training iterations, the pseudo-loss tends to converge to a similar level. Although the pseudo-loss provides a precise indication of a sample's representativeness in the training set, obtaining accurate pseudo labels can be challenging, requiring careful selection of hyperparameters such as the number of training steps, early stopping, and etc. Otherwise, inaccurate pseudo-labels can mislead the selection of unseen samples in AL.

### 5.3 Domain-Invariant Active Learning

**Tracking training trajectory.** Based on the above observations, we proposed an active learning strategy DIAL, outlined in Algorithm 1. Unlike other AL methods, which typically rely on the inference from a single trained model $f_{\mathcal{D}_L}$, DIAL leverages the complete training trajectory in the acquisition process. Prior to the acquisition step, DIAL tracks the trajectory by collecting checkpoints $\mathcal{M} = \{f_{\theta_1}, \dots, f_{\theta_T} \mid \mathcal{D}_L\}$ throughout the training. This can be viewed as discretizing the evolution of the model over time $f_{\theta(t)}$, akin to approximating the quantity $\partial f(x)/\partial t$. In practical implementation, DIAL does not track every single step but instead samples a fixed number of checkpoints evenly across the training process. This enables us to capture the differences in learning dynamics between overrepresented and underrepresented samples, a characteristic that naturally emerges in the standard training procedure. Our empirical findings validate the efficacy of this approach, particularly under biased scenarios.

**Algorithm 1** Domain-Invariant Active Learning
___
**Input:** number of gradient update steps $T$, number of acquisitions $k$
> **for** $n \in [N]$ **do**
>> ▷ *Train $f_\theta$ and collect checkpoints* ◁
>> $\mathcal{M} \leftarrow \{\}$
>> **for** $t \in [T]$ **do**
>>> $\theta_{t+1} \leftarrow \theta_t + \eta \nabla_{\theta_t} \ell(\mathcal{D}_L)$
>>> $\mathcal{M} \leftarrow \mathcal{M} \cup \{\theta_{t+1}\}$
>>
>> ▷ *acquire new samples* ◁
>> **if** $n < N$ **then**
>>> compute $d_\mathcal{M}(x)$ for all $x \in \mathcal{D}_U$
>>> acquire $Q = \{\text{top-k } d_\mathcal{M}\}$
>>> $\mathcal{D}_L \leftarrow \mathcal{D}_L \cup \mathcal{H}(Q) \ ; \ \mathcal{D}_U \leftarrow \mathcal{D}_U \setminus Q$
>
> **output** $f_{\mathcal{D}_L}$

**Acquisition function.** After collecting the trajectory of training $\mathcal{M}$, DIAL computes the informativeness of $x \in \mathcal{D}_U$ using $d_\mathcal{M}$. Essentially, $d_\mathcal{M}$ computes the pairwise disparity of predictions between each $\theta \in \mathcal{M}$

$$d_\mathcal{M}(x) = \sum_{\theta' \in \mathcal{M}} \sum_{\theta \in \mathcal{M}} \phi(f_\theta(x), f_{\theta'}(x)) \qquad (4)$$

where $\phi$ is the distance function measuring the discrepancy between the outputs from two different models. A high value of $d_\mathcal{M}(x)$ indicates a slow evolution of predictions for $x$ during training, suggesting that $x$ is either a challenging sample or belongs to a underrepresented group. We also explored some existing ensemble-based acquisition functions in our experiments (refer to Section 7) since the set of trajectories $\mathcal{M}$ can be viewed as an ensemble of the model.

**Working principle.** The working principle of DIAL is visually detailed in Figure 6. Predictions for individual samples across various subgroups by different checkpoints in the trajectory $\mathcal{M} = \{\theta_1, \ldots, \theta_T\}$ are mapped onto the probability simplex located at the top left of Figure 6. Here, it can be observed that the predictions associated with BA samples typically tend to aggregate towards the vertices of the simplex. In contrast, predictions of BC samples exhibit a more dispersed pattern, spanning more uniformly across the simplex. This variation in distribution across the simplex indicates a substantial difference in prediction consistency between checkpoints, especially noticeable among the BC samples. This phenomenon is further quantified through the pairwise distances matrix, where distances are calculated with metric $\phi(f_\theta(x), f_{\theta'}(x))$. The distance matrices reveal greater discrepancies in the predictions over successive training checkpoints for BC samples compared to BA samples, suggesting less consistency in the model's performance across different training stages for BC samples. In light of this observations, it becomes a useful strategy to enhance the balancedness of $\mathcal{D}_L$. By selecting samples $x \in \mathcal{D}_U$ that demonstrate high values of $d_\mathcal{M}$, it is likely to include more samples from underrepresented

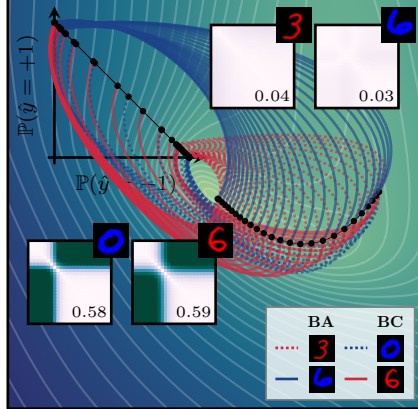

Figure 6: Illustration of DIAL alongside pairwise distance matrices for samples from each subgroup. Average values of distance matrices are shown in the corner of the matrices.

subgroups into $\mathcal{D}_L$. Such targeted acquisition is instrumental in mitigating spurious correlations present in the existing $\mathcal{D}_L$. Ultimately, this approach leads to the development of a model that is more robust and better attuned to variations w.r.t. the spurious attribute.

**DIAL acquires more balanced underrepresented batches.** In Figure 7, we can see that the distributions for the BC subgroups are long-tailed, skewing towards higher informativeness. This pattern is especially pronounced for DIAL, where the distributions for both BC subgroups are more symmetric. Conversely, other methods like Margin and Entropy, while capturing the informativeness of the BC subgroups, produce distributions that are less symmetric. For instance, the BC subgroup ($y = +1, s = $ red) demonstrates a higher density towards the upper score range compared to its counterpart ($y = -1, s = $ blue). This disparity in distribution suggests bias in the acquisition process, where one underrepresented subgroup will be disproportionately represented over the other.[4] Given this context, when adopting a greedy approach

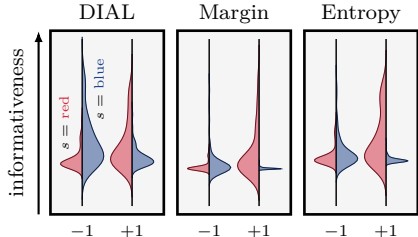

Figure 7: Distributions of informativeness across subgroups with acquisition functions.

___
[4]This might result improvements in certain underrepresented subgroups, while others will still remain underrepresented, leading to only marginal improvements in worst-group accuracy.

to sample selection (i.e., maximize the informativeness of each batch of samples), DIAL demonstrates a superior capability to ensure a more balanced inclusion of samples from underrepresented subgroups. Moreover, this implies that DIAL can achieve better group-wise performance, such as improved worst-group accuracy.

## 6 Experiments

In this section, we provide empirical evaluations on various benchmarks along with several baseline methods. The results are averaged across five trials conducted with different random seeds. Mean and standard deviation are presented in all figures and tables. All experimental details can be found in Appendix A.

**Datasets and DNNs.** We assess the performance of all AL algorithms using two categories of benchmark datasets. The first category is datasets with spurious correlations for evaluating subgroup robustness, including Waterbirds (Sagawa et al., 2020), CelebA (Liu et al., 2015) and Corrupted CIFAR-10 (Hendrycks & Dietterich, 2019). Waterbirds is the classification task for landbird vs. waterbird where samples are associated with different backgrounds (spurious attribute): land or water; in the training set landbirds (or waterbirds) are more likely to appear on land (or on water). In the CelebA dataset, the task is to determine if the person in the image has black hair, where gender is considered the spurious attribute. The dataset predominantly consists of females without black hair. Corrupted CIFAR-10 are construed using the standard CIFAR-10 dataset with several types of textures applied onto the images where some classes are entangled with some specific textures in the training set. The second category is the common AL benchmark datasets: CIFAR-10 (Krizhevsky & Hinton, 2009) and SVHN (Netzer et al., 2011) for assessing the performance of DIAL in the unbiased environment. Further details regarding the datasets are provided in Appendix A.1. In terms of network architectures, we employ ResNet50 and ResNet18 (He et al., 2016). Comprehensive information such as training hyperparameters can be found in Appendix A.2.

**Baselines.** In addition to Random sampling, which selects samples randomly to simulate passive learning, we include the following baseline methods: BADGE (Ash et al., 2020), CoreSet (Sener & Savarese, 2018), Margin (Roth & Small, 2006), Confidence (Wang & Shang, 2014), Entropy (Wang & Shang, 2014), BAIT (Ash et al., 2021) and Cluster-Margin (Citovsky et al., 2021). Further details on these baseline methods can be found in Appendix A.3.

**Metrics.** Regarding performance evaluation, we use worst-group accuracy as the primary metric for Waterbirds and CelebA, and average accuracy for Corrupted CIFAR-10 (as suggested by Nam et al. (2020), given the unbiased test split). While for all common AL datasets, we rely on the average accuracy. Additionally, we also incorporate the area under learning curve (ALC) metric, which quantifies the area under the respective metric curve (y-axis) across the number of queried batches (x-axis).

### 6.1 Results

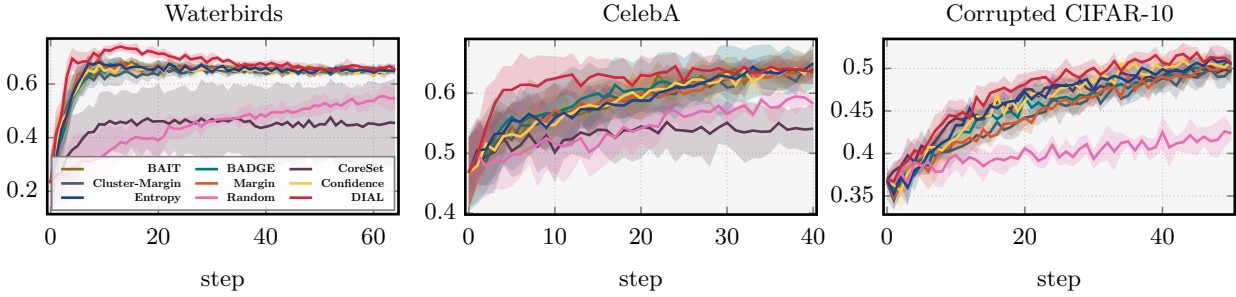

Figure 8: Performance vs. number of AL steps. The figures display worst-group accuracy for Waterbirds and CelebA, alongside average accuracy for Corrupted CIFAR-10.

**Does AL improve subgroup performance?** Figure 8 presents the performance achieved by various AL algorithms. With the exception of CoreSet on Waterbirds, all baseline methods consistently outperform passive learning (Random) throughout the run. This observation indicates that AL methods inherently possess the capability to enhance subgroup robustness in a sample-efficient manner, validating our research hypothesis. Particularly noteworthy is the performance of DIAL, which consistently outperforms all baseline methods, especially on Waterbirds. At its peak performance, DIAL achieves approximately 74% accuracy with only around 13% of the total training samples, surpassing state of the art (SOTA) algorithms such as those by Sagawa et al. (2020) (73.1%) and Liu et al. (2021) (71.2%)[5]. A comprehensive comparison with the SOTA methods is provided in Appendix B.4.3.

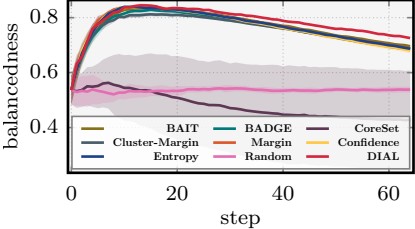

Figure 9: Balancedness[6]of $\mathcal{D}_L$ on Waterbirds throughout the run. Higher implies that $\mathcal{D}_L$ is more balanced, i.e., a value of 1 means that the numbers of samples from every subgroups $g \in \mathcal{G}$ are equal.

While uncertainty methods demonstrate competitive performance, diversity-based approaches (BADGE, CoreSet and Cluster-Margin) exhibit less potency. This discrepancy can be attributed to the enforced diversity criteria, which result in a certain proportion of queried samples being drawn from overrepresented subgroups, thereby failing to adequately address spurious correlations in the labelled pool. It is important to note the decreasing trend observed for DIAL after reaching peak performance on Waterbirds, which is attributed to the insufficient underrepresented samples in $\mathcal{D}_U$ (see Figure 10a), exacerbating spurious correlations upon continued sample acquisition. Consequently, we use relatively small query batch sizes to prevent excessive consumption of underrepresented samples, ensuring optimal performance (further analysis is provided in Section 7).

Additionally, Figure 9 illustrates the balancedness of $\mathcal{D}_L$, quantified by the normalized entropy of subgroup proportions: $\sum_{g \in \mathcal{G}} -p_g \log p_g / \log |\mathcal{G}|$, where $p_g$ represents the proportion of subgroup $g$. Due to space limit, the balancedness for other datasets are presented in Appendix B. The analysis reveals a correlation between balancedness and subgroup performance, as observed in Figure 8: higher balancedness in $\mathcal{D}_L$ corresponds to improved robustness. The performance trend of Random remains consistent throughout, indicating that performance gains over AL steps stem primarily from the overall growth of the labelled set rather than the acquisition of underrepresented samples. During the early steps, when the balancedness achieved by DIAL is comparable to that of other methods, the subgroup performance of DIAL aligns closely with that of its counterparts. However, as DIAL approaches peak balancedness, subgroup performance reaches its zenith. This observation reinforces our core objective of designing a subgroup-robust AL method: acquiring new samples to balance the training pool, thereby achieving optimal robustness.

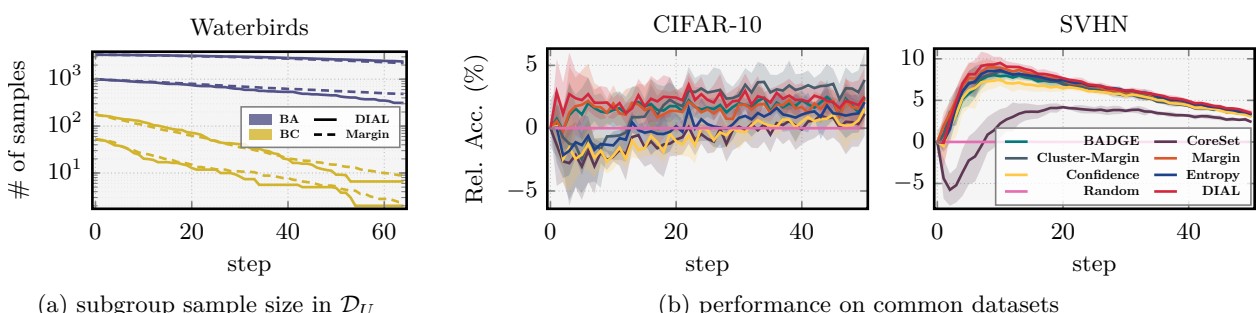

(a) subgroup sample size in $\mathcal{D}_U$          (b) performance on common datasets

Figure 10: (a) Evolution of $\mathcal{D}_U$ on Waterbirds. Each line represents an individual subgroup. Due to the bias, there are substantially more BA subgroup samples initially. Observably, DIAL tends to "consume" a greater proportion of BC samples, extending to even the secondary subgroup within BA. (b) Performance vs. number of AL steps for common datasets. The y-axis represents the relative improvement (percentage change) in comparison to Random.

---

[5]The numerical values are directly taken from Yang et al. (2023)
[6]The curves of Margin and Entropy overlap.

**Standard AL setting.** Figure 10b shows the performance curves for the standard datasets. DIAL exhibits competitive performance on both datasets, with its advantage being particularly pronounced on SVHN dataset. This observation underscores the versatility of DIAL: not only does it excel in imbalanced settings, but it also proves effective in conventional settings.

| | First 10 AL steps | | | | | | Full run | | | | | |
|---|---|---|---|---|---|---|---|---|---|---|---|---|
| Algorithm | Waterbirds | CelebA | Co. CIFAR-10 | SVHN | CIFAR-10 | Rank | Waterbirds | CelebA | Co. CIFAR-10 | SVHN | CIFAR-10 | Rank |
| Random | $2.89_{\pm 1.0}$ | $4.99_{\pm 0.5}$ | $3.81_{\pm 0.1}$ | $5.89_{\pm 0.1}$ | $4.64_{\pm 0.1}$ | $6.4_{\pm 2.0}$ | $28.36_{\pm 2.6}$ | $21.66_{\pm 1.1}$ | $20.00_{\pm 0.4}$ | $39.58_{\pm 0.2}$ | $28.62_{\pm 0.1}$ | $5.6_{\pm 2.0}$ |
| Margin | $\mathbf{5.54}_{\pm 0.4}$ | $\mathbf{5.39}_{\pm 0.3}$ | $3.83_{\pm 0.1}$ | $\mathbf{6.30}_{\pm 0.1}$ | $\mathbf{4.70}_{\pm 0.1}$ | $5.8_{\pm 1.6}$ | $\mathbf{41.02}_{\pm 0.6}$ | $23.58_{\pm 1.0}$ | $22.38_{\pm 0.2}$ | $\mathbf{42.00}_{\pm 0.2}$ | $29.06_{\pm 0.3}$ | $5.6_{\pm 2.3}$ |
| Certainty | $5.47_{\pm 0.3}$ | $5.23_{\pm 0.2}$ | $3.89_{\pm 0.1}$ | $6.21_{\pm 0.1}$ | $4.56_{\pm 0.0}$ | $4.4_{\pm 1.6}$ | $40.72_{\pm 0.4}$ | $23.50_{\pm 0.7}$ | $23.10_{\pm 0.3}$ | $41.71_{\pm 0.2}$ | $28.48_{\pm 0.1}$ | $5.0_{\pm 2.5}$ |
| Entropy | $5.51_{\pm 0.5}$ | $5.36_{\pm 0.3}$ | $3.89_{\pm 0.1}$ | $6.26_{\pm 0.1}$ | $4.58_{\pm 0.0}$ | $\mathbf{2.4}_{\pm 0.5}$ | $40.92_{\pm 0.6}$ | $23.57_{\pm 0.9}$ | $23.10_{\pm 0.2}$ | $41.97_{\pm 0.2}$ | $28.70_{\pm 0.2}$ | $\mathbf{3.8}_{\pm 1.5}$ |
| CoreSet | $3.61_{\pm 1.0}$ | $5.08_{\pm 0.4}$ | $\mathbf{4.05}_{\pm 0.1}$ | $5.79_{\pm 0.1}$ | $4.53_{\pm 0.1}$ | $5.8_{\pm 1.2}$ | $28.20_{\pm 8.1}$ | $21.20_{\pm 1.5}$ | $\mathbf{23.17}_{\pm 0.3}$ | $40.64_{\pm 0.2}$ | $28.45_{\pm 0.2}$ | $5.2_{\pm 1.3}$ |
| BAIT | $5.44_{\pm 0.5}$ | $5.30_{\pm 0.3}$ | – | – | – | – | $40.94_{\pm 0.6}$ | $23.61_{\pm 0.9}$ | – | – | – | – |
| BADGE | $5.32_{\pm 0.4}$ | $5.37_{\pm 0.5}$ | $3.87_{\pm 0.1}$ | $6.23_{\pm 0.1}$ | $4.69_{\pm 0.0}$ | $3.8_{\pm 1.5}$ | $40.69_{\pm 0.4}$ | $\mathbf{23.83}_{\pm 1.9}$ | $22.68_{\pm 0.3}$ | $41.86_{\pm 0.2}$ | $29.10_{\pm 0.2}$ | $5.2_{\pm 1.3}$ |
| Cluster-Margin | $5.36_{\pm 0.4}$ | $5.27_{\pm 0.5}$ | $3.85_{\pm 0.1}$ | $6.26_{\pm 0.1}$ | $4.59_{\pm 0.1}$ | $6.4_{\pm 1.2}$ | $40.35_{\pm 0.8}$ | $23.48_{\pm 1.2}$ | $22.31_{\pm 0.3}$ | $41.84_{\pm 0.2}$ | $\mathbf{29.18}_{\pm 0.4}$ | $4.6_{\pm 2.1}$ |
| DIAL | $\mathbf{6.16}_{\pm 0.4}$ | $\mathbf{5.84}_{\pm 0.5}$ | $\mathbf{4.07}_{\pm 0.1}$ | $\mathbf{6.30}_{\pm 0.1}$ | $\mathbf{4.74}_{\pm 0.0}$ | $\mathbf{1.0}_{\pm 0.0}$ | $\mathbf{43.10}_{\pm 0.6}$ | $\mathbf{24.83}_{\pm 1.0}$ | $\mathbf{23.79}_{\pm 0.3}$ | $\mathbf{42.09}_{\pm 0.1}$ | $\mathbf{29.25}_{\pm 0.1}$ | $\mathbf{1.0}_{\pm 0.0}$ |

Table 1: The ALC for all datasets with their corresponding metrics. The left section of the table shows the ALC over the initial 10 AL steps, while the right section displays the ALC for the entire run. The highest and second-highest values are highlighted in **bold red** and **bold blue**, respectively.

Table 1 provides a comprehensive numerical overview of the performance of all methods. Throughout the early stages and the entire run, DIAL consistently outperforms all baseline methods across all datasets. Following closely, uncertainty-based methods exhibit the second-best performance on all datasets except for Corrupted CIFAR-10 (and CIFAR-10 in full run). Margin and Entropy methods, on average, demonstrate competitive performance. Furthermore, while DIAL may not exhibit notably superior performance on SVHN, it remains comparable to baseline methods, indicating its applicability to problems where imbalance or spurious correlations are not present, without compromising performance. Refer to Appendix B for comprehensive summaries of other metrics per dataset, similar to those presented in Yang et al. (2023).

# 7 Ablation study

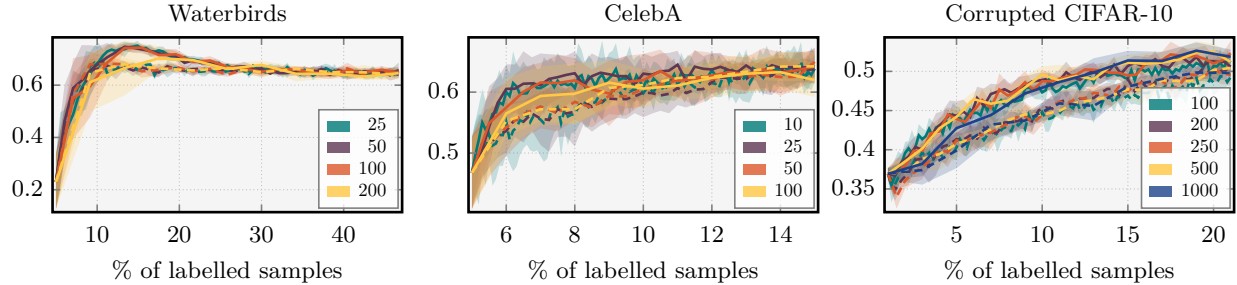

Figure 11: Learning curve (worst-group accuracy for Waterbirds and CelebA, and average accuracy for Corrupted CIFAR-10) across different query batch sizes. The solid line and dashed line represent DIAL and Margin, respectively.

**Effect of query batch size** Figure 11 presents the subgroup robustness performance w.r.t. the query batch size $k$. Similar to standard AL outcomes, the subgroup performance tends to decline as $k$ increases. This decline occurs because larger values of $k$ compel the querying of uninformative samples, which, in this context, correspond to overrepresented or BC samples. Remarkably, with a batch size of 25, DIAL achieved 76.1% on Waterbirds, outperforming most passive baselines as reported in Yang et al. (2023). Thus, when dealing with imbalance in AL, it is recommended to use smaller query batch sizes. Additionally, it is noteworthy that DIAL consistently outperforms Margin, which is the best-performing baseline overall, across all query batch sizes.

**Effect of $|\mathcal{M}|$** Figure 12 depicts the learning curve across different densities of learning dynamics for DIAL. We uniformly sample model checkpoints throughout the training process. We observed that a larger size of $\mathcal{M}$ (denser sampling) generally leads to higher performance, as it provides a more accurate estimation of

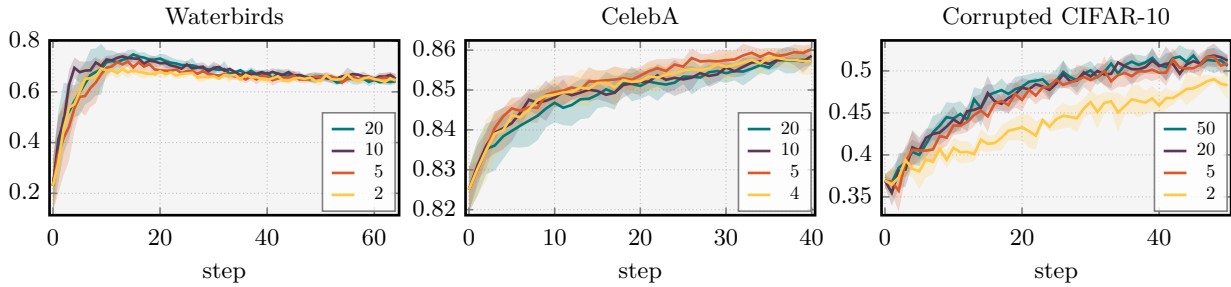

Figure 12: Learning curve (worst-group accuracy) across different size of checkpoints $|\mathcal{M}|$.

the learning dynamics. In the case of Waterbirds, the slow increase in performance in the beginning when using $|\mathcal{M}| = 20$ could be attributed to noise caused training instability. However, despite this, it achieved higher peak performance compared to other configurations. Overall, it's crucial to strike a balance in the sampling rate. Sampling rates should not be excessively high (resulting in larger $\mathcal{M}$), as this may capture training noise. Conversely, they should not be too low (resulting in smaller $\mathcal{M}$), as this could lead to missing information of learning dynamics. In the experiments presented in Figure 8, we used $|\mathcal{M}| = 10$ for Waterbirds, CelebA, CIFAR-10, and SVHN, while using $|\mathcal{M}| = 50$ for Corrupted CIFAR-10.

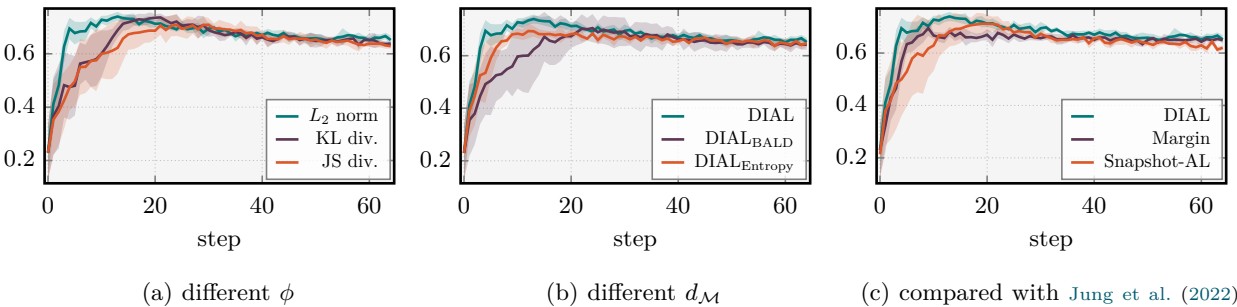

(a) different $\phi$                (b) different $d_{\mathcal{M}}$                (c) compared with Jung et al. (2022)

Figure 13: Performance of DIAL (worst-group accuracy) across different configurations evaluated on Waterbirds. (a) Evaluation of various distance functions $\phi$. (b) Assessment of different common acquisition criteria on training dynamics $d_{\mathcal{M}}$. (c) Assessment of the approach proposed by Jung et al. (2022).

**Effect of distance functions $\phi$.** We conducted a comparison of distance functions utilized to capture predictive discrepancies among checkpoints. The discrepancy is evaluated using the softmax output and we found that this approach resulted in much better performance, even with $L_2$ norm. The comparison in Figure 12a shows that both the $L_2$ norm and Kullback-Leibler (KL) divergence demonstrate similar peak performance. However, the $L_2$ norm achieves this with significantly fewer acquisition steps. Jensen-Shannon (JS) divergence, a symmetric version of the KL divergence, initially behaves similarly to the KL divergence but diverges after approximately 10 acquisition steps, eventually converging to a comparable level. Overall, the $L_2$ norm proves adept at accurately detecting predictive discrepancies, particularly on underrepresented samples, leading to rapid improvements in robustness. The diminished performance with divergence metrics may be attributed to the non-linear operation, which tends to neglect very small discrepancies. We used the $L_2$ norm for DIAL in all the experiments.

**Effect of discrepancy criteria $d_{\mathcal{M}}$.** In addition to Equation (4), we also explored alternative criteria for assessing informativeness with training dynamics $\mathcal{M}$. One such criteria is Entropy, defined as $d_{\mathcal{M}}(x) = H(f_{\mathcal{M}}(x))$, where $f_{\mathcal{M}}(x) = \frac{1}{|\mathcal{M}|} \sum_{\theta \in \mathcal{M}} f_\theta(x)$ and H is the entropy measure which evaluates the overall predictive entropy of a sample $x$ throughout the training. Additionally, we also incorporate BALD (Houlsby et al., 2011) $d_{\mathcal{M}}(x) = H(f_{\mathcal{M}}(x)) + \frac{1}{|\mathcal{M}|} \sum_{\theta \in \mathcal{M}} H(f_\theta(x))$ to measure the predictive disagreement among models throughout the training. Figure 12b shows that both variants ($\text{DIAL}_{\text{Entropy}}$ and $\text{DIAL}_{\text{BALD}}$) do not yield performance gain compared to the proposed criteria (in Equation (4)).

**Comparison with Snapshot-AL** We conducted a comparison between DIAL and Snapshot-AL as proposed by Jung et al. (2022). To ensure a fair comparison, we obtained an equal number of checkpoints for both DIAL and Snapshot-AL, and employed the same disagreement criteria as outlined in Equation (4). The primary distinction lies in the application of the cosine annealing learning rate scheduler (Loshchilov & Hutter, 2017) during training with Snapshot-AL. As illustrated in Figure 12c, Snapshot-AL, which utilizes diverse checkpoints generated with the learning rate scheduler, consistently underperforms DIAL throughout run. This further corroborates our assertion that the diversity of ensembles or checkpoints does not effectively facilitate the identification of underrepresented samples. Rather, tracking the learning dynamics provides a more accurate means of identification of such samples. Nevertheless, Snapshot-AL demonstrates superior peak performance compared to Margin, despite its relatively lower performance in the early rounds.

## 8 Conclusion, limitations and future work

In this work, we investigate spurious correlations within in the AL settings. We demonstrate a potential failure mode of uncertainty-based AL approaches in handling spurious correlations. To address this, we propose a simple AL method that effectively identifies underrepresented samples from the unlabelled pool. Our experiments on real-world datasets show that our proposed method mitigates bias in the labelled dataset and outperforms baseline methods. Additionally, we provide a detailed analysis highlighting the effectiveness of our approach with respect to various AL hyperparameters.

Despite demonstrating promising results, DIAL has some limitations regarding the computational complexity. The time complexity for the forward pass of DIAL is $\mathcal{O}(mn)$, where $m = |\mathcal{M}|$ is the number of checkpoints, and $n = |\mathcal{D}_U|$ is the number of unlabelled samples, as opposed to $\mathcal{O}(n)$ for simpler methods like Margin. The acquisition function of DIAL compares the predictions $f_\theta(x)$ between checkpoints $(\theta, \theta') \in \mathcal{M} \times \mathcal{M}$, resulting in an overall time complexity of $\mathcal{O}(mn(1 + m))$, requiring $(m + m^2)$ more computations compared to Margin. Moreover, DIAL only computes acquisition scores based on the model outputs $f_\theta(x)$, which correspond to the number of classes and are typically much smaller than high-dimensional feature embeddings. Consequently, when the dimension of the model output is small, the complexity is primarily dominated by the forward passes of the checkpoints, resulting in an approximate complexity of $\mathcal{O}(mn)$. However, as shown in our ablation study, $m$ does not need to be extremely large to achieve good performance. Thus, the computational complexity of DIAL remains manageable in practical applications.

Future work could explore techniques to reduce this computational burden, such as using approximation methods with fewer checkpoints to estimate prediction deviations along the training trajectory. Furthermore, DIAL's runtime scales linearly with the number of samples (i.e., $\mathcal{O}(n)$), making it far more efficient than methods that rely on costly matrix operations, such as matrix inversion or multiplication with the kernel matrix (Ash et al., 2021), which can scale up to $\mathcal{O}(n^3)$. Another limitation of DIAL is that it requires a pre-defined number of checkpoints to be sampled during training, which can be challenging to determine in practice.

Additionally, in the current framework, all acquired samples in the labelled pool are treated with equal importance as the existing ones. One potential direction for future work is to develop a reweighting mechanism alongside the AL acquisition process to assign greater importance to the acquired underrepresented samples for better correcting the bias in the labelled pool. Additionally, future research could explore the use of foundation models to enhance performance and accelerate the AL process (Gupte et al., 2024).

**Broader Impact Statement**

In this work, we introduced DIAL, an AL method which makes more efficient use of data by focusing on areas of imbalance particularly when dealing with underrepresented samples. This approach not only improves the accuracy of models but also promotes fairness and equity by ensuring that marginalised or rare cases are properly represented. This is particularly crucial in domains where biased or erroneous predictions can have serious consequences, such as predicting medical outcomes or making financial decisions. By mitigating spurious correlations, we also reduce the risk of overfitting models to patterns that do not hold across diverse populations, thereby improving their generalization to real-world data. Furthermore, as AL methods improve

model efficiency, organisations can make more informed decisions with less data, reducing the environmental costs of large-scale data collection and processing. Thus, our work not only improves model performance and fairness but also contributes to sustainable AI practices.

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

# A  Experimental setup

## A.1  Datasets

In all experiments, we use the training split as the unlabelled pool and evaluate the performance using the test split. At the start of the AL loops, a subset of samples is randomly labelled to form the initial labelled pool. Dataset statistics are provided in Figure A.1.

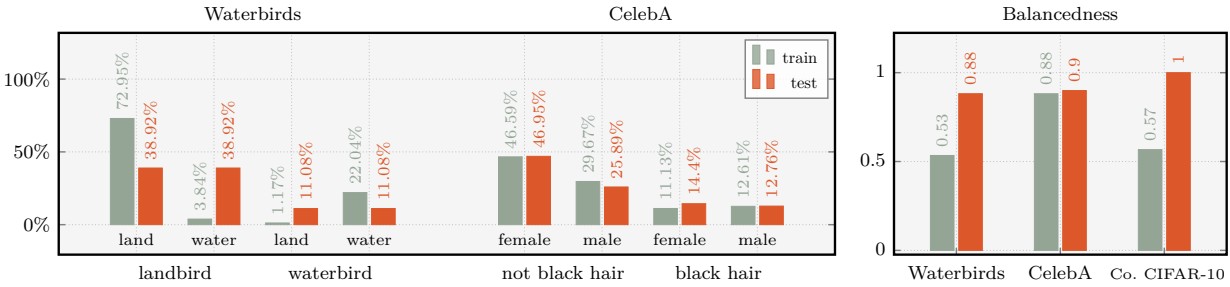

Figure A.1: **(left)** The subgroup distribution of the training and the test splits over all subgroups. **(right)** The balancedness of the dataset.

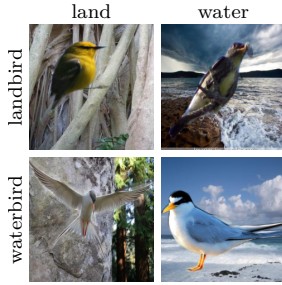

**Waterbirds.**  The Waterbirds dataset is a synthetic dataset created by overlaying bird images from Wah et al. (2011) onto various scene images from Zhou et al. (2018). The target labels are *landbird* and *waterbird*, while the spurious attribute is the background: *land* or *water*. In the training set, there is a spurious correlation between the bird class and the background, with the majority subgroups (or BA subgroups) being *landbirds on land* and *waterbirds on water*. Both the target and spurious labels are binary: $\mathcal{Y} = $ landbird, waterbird and $\mathcal{S} = $ land, water, resulting in four subgroups ($|\mathcal{G}| = 4$). The train and test splits consist of 4,795 and 5,794 samples, respectively.

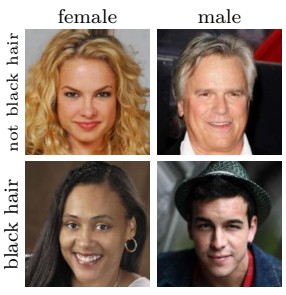

**CelebA.**  The CelebA dataset (Liu et al., 2015) comprises over 200,000 celebrity images annotated with 40 different attributes. In our experiment, we use the `Black_Hair` attribute as the target and the `Male` attribute as the spurious attribute. The dataset exhibits a spurious correlation between hair colour and gender; specifically, with most females having non-black hair. Both target and spurious labels are binary: $\mathcal{Y} = $ not black hair, black hair and $\mathcal{S} = $ female, male, resulting in four subgroups ($|\mathcal{G}| = 4$). Due to computational constraints, we randomly subsampled 10,000 samples from the training split and used the original test split, which contains 19,962 samples.

**Corrupted CIFAR-10.**  The Corrupted CIFAR-10 dataset is generated from the standard CIFAR-10 (Krizhevsky & Hinton, 2009) by applying corruptions as proposed by Hendrycks & Dietterich (2019). The target labels are the original CIFAR-10 labels, while the spurious labels correspond to the types of corruption: (1) Gaussian Noise, (2) Shot Noise, (3) Impulse Noise, (4) Speckle Noise, (5) Gaussian Blur, (6) Defocus Blur, (7) Glass Blur, (8) Motion Blur, (9) Zoom Blur, (10) Original. Following the procedure defined by Nam et al. (2020), we generate the dataset with a corruption severity of 1 and a ratio of 95% BA samples to create the spurious correlation. Subgroups where the numeric index of target label and that of spurious label coincide ($y = s$) are considered BA subgroups, while others are BC subgroups. Both target and spurious labels have a cardinality of 10, resulting in 100 subgroups ($|\mathcal{G}| = 100$). The subgroups are uniformly distributed, with BA and BC subgroups having proportions of 95%/10 and 5%/90, respectively. The train and test splits consist of 50,000 and 10,000 samples, respectively, with the test split being unbiased (uniformly distributed across all subgroups).

## A.2 Implementation details and hyperparameter settings

Our code for the experiments is based on the Baal AL framework (Atighehchian et al., 2022). We use Baal implementations for Random, Margin, and Entropy, while BADGE, Coreset and BAIT are sourced from the repository by Ash et al. (2020), and Cluster-Margin is sourced from the unofficial implementation. We construct the Waterbirds and CelebA datasets using the WILDS package (Koh et al., 2021) and the Corrupted CIFAR-10 dataset using the repository by Nam et al. (2020). For SOTA and robustness evaluation, we use the code by Yang et al. (2023). Additionally, for visualizing the training trajectories in Section 5, we deploy the loss-landscape-anim repository. We also followed some hyperparameters setting and data preprocessing procedure specified in Yang et al. (2023) and Nam et al. (2020). The code for the experiments in this work is available at `https://anonymous.4open.science/r/jSYaklvh`, and it will be made publicly available upon acceptance.

| Dataset | # of initial samples | query batch size | architecture | pre-trained weights | optimizer | learning rate | weight decay | momentum | batch size |
|---|---|---|---|---|---|---|---|---|---|
| Waterbirds | 5% of total | 0.625% of total | ResNet50 | ImageNet-1K | SGD | $1 \times 10^{-3}$ | $1 \times 10^{-4}$ | 0.9 | 108 |
| CelebA | 5% of total | 0.25% of total | ResNet50 | ImageNet-1K | SGD | $1 \times 10^{-3}$ | $1 \times 10^{-4}$ | 0.9 | 108 |
| Corrupted CIFAR-10 | 500 | 200 | ResNet18 | ImageNet-1K | Adam | $1 \times 10^{-3}$ | $1 \times 10^{-4}$ | – | 128 |
| CIFAR-10 | 500 | 500 | ResNet18 | – | SGD | $1 \times 10^{-3}$ | $5 \times 10^{-4}$ | 0.9 | 128 |
| SVHN | 500 | 500 | ResNet18 | ImageNet-1K | SGD | $1 \times 10^{-3}$ | $1 \times 10^{-4}$ | 0.9 | 128 |

## A.3 Baselines

Random — Uniformly label $k$ samples from the unlabelled pool, resembling the passive setting.

Margin — An uncertainty-based AL method that selects samples based on the difference between the top two predictive class probabilities: $\arg\max_{x \in \mathcal{D}_U} f(x)^0 - f(x)^1$, where $f(x)^0$ and $f(x)^1$ are the largest and second largest softmax entries of $f(x)$ (Roth & Small, 2006).

Entropy — An uncertainty-based AL method that selects the most uncertain samples based on predictive entropy: $\arg\max_{x \in \mathcal{D}_U} H(f(x))$, where $H$ is the entropy defined as $H(p) = \sum_{i=1}^{C} -p_i \log(1/p_i)$ (Wang & Shang, 2014).

Confidence — An uncertainty-based AL method that selects samples with the smallest predictive probability: $\arg\min_{x \in \mathcal{D}_U} \max_{i \in [C]} f(x)_i$ (Wang & Shang, 2014).

BADGE — A diversity-based AL method. For each unlabelled sample, it computes the gradient embeddings (the first derivative of the pseudo-loss with respect to the output of the penultimate layer). It then selects a subset of $k$ samples using the k-MEANS++ seeding algorithm on the gradient embeddings (Ash et al., 2020).

Coreset — A diversity-based AL method that selects a subset of $k$ samples by solving the coreset problem on the feature embeddings (the output of the penultimate layer) (Sener & Savarese, 2018).

BAIT — A neural AL method that selects samples by minimising the expected risk, estimated using Fisher information, which is computed using gradient embeddings (Ash et al., 2021).

Cluster-Margin — An AL method that selects samples by balancing exploration (diversity) and exploitation (uncertainty). It first perform clustering on $\mathcal{D}_U \cup \mathcal{D}_L$ and then selects samples based on the margin criterion within each cluster (Citovsky et al., 2021).

# B Additional results

In this section, we present additional results and analysis on datasets with spurious correlations. We include the ALC over evaluation metrics as presented in Yang et al. (2023) (refer to Section B.3 in Yang et al. (2023) for details). Additionally, we report the *robust accuracy*, defined as the *worst-case expected accuracy over attribute* (Sohoni et al., 2020): $\min_{s' \in \mathcal{S}} \mathbb{E}_{(x,y)|s=s'} [\mathbb{1}[f(x) = y]]$.

## B.1 Waterbirds

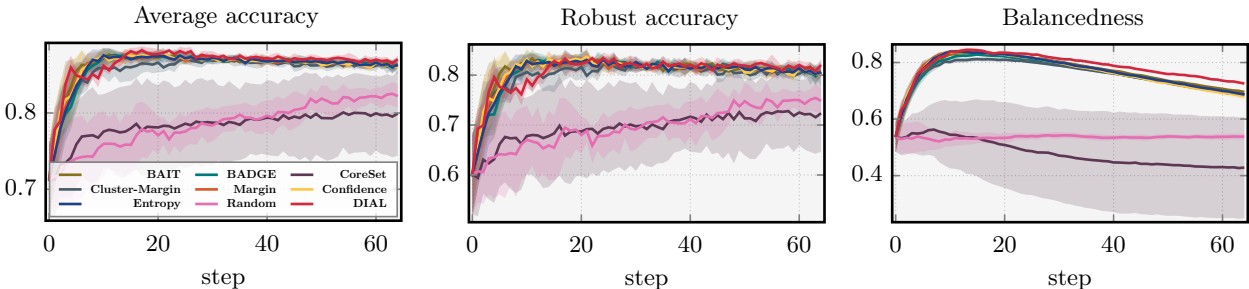

### B.1.1 ALC over evaluation metrics across different AL methods

| Algorithm | Avg Acc. | Worst Acc. | Avg Prec. | Worst Prec. | Avg F1 | Worst F1 | Adjusted Acc. | Balanced Acc. | AUROC | ECE |
|---|---|---|---|---|---|---|---|---|---|---|
| Random | $50.2_{\pm 1.4}$ | $28.4_{\pm 2.6}$ | $45.3_{\pm 1.4}$ | $33.0_{\pm 2.3}$ | $46.2_{\pm 1.5}$ | $37.7_{\pm 2.0}$ | $48.0_{\pm 1.4}$ | $48.0_{\pm 1.4}$ | $54.5_{\pm 1.2}$ | $9.7_{\pm 1.4}$ |
| Margin | $55.2_{\pm 0.3}$ | $41.0_{\pm 0.6}$ | $51.1_{\pm 0.4}$ | $42.2_{\pm 0.8}$ | $52.1_{\pm 0.4}$ | $46.1_{\pm 0.5}$ | $53.6_{\pm 0.3}$ | $53.6_{\pm 0.3}$ | $58.9_{\pm 0.2}$ | $6.1_{\pm 0.3}$ |
| Confidence | $55.2_{\pm 0.4}$ | $40.7_{\pm 0.4}$ | $51.1_{\pm 0.4}$ | $42.3_{\pm 0.8}$ | $52.1_{\pm 0.4}$ | $46.0_{\pm 0.5}$ | $53.6_{\pm 0.3}$ | $53.6_{\pm 0.3}$ | $58.9_{\pm 0.2}$ | $6.1_{\pm 0.3}$ |
| Entropy | $55.2_{\pm 0.4}$ | $40.9_{\pm 0.6}$ | $51.0_{\pm 0.5}$ | $42.1_{\pm 0.8}$ | $52.1_{\pm 0.4}$ | $46.0_{\pm 0.6}$ | $53.6_{\pm 0.3}$ | $53.6_{\pm 0.3}$ | $58.9_{\pm 0.2}$ | $6.1_{\pm 0.3}$ |
| CoreSet | $50.1_{\pm 3.4}$ | $28.2_{\pm 8.1}$ | $45.3_{\pm 3.5}$ | $33.0_{\pm 5.6}$ | $46.0_{\pm 3.8}$ | $37.5_{\pm 5.2}$ | $47.8_{\pm 3.6}$ | $47.8_{\pm 3.6}$ | $54.2_{\pm 3.0}$ | $9.4_{\pm 2.5}$ |
| BAIT | $55.3_{\pm 0.3}$ | $40.9_{\pm 0.6}$ | $51.2_{\pm 0.4}$ | $42.3_{\pm 0.7}$ | $52.2_{\pm 0.4}$ | $46.1_{\pm 0.5}$ | $53.6_{\pm 0.3}$ | $53.6_{\pm 0.3}$ | $58.9_{\pm 0.2}$ | $6.0_{\pm 0.3}$ |
| BADGE | $55.1_{\pm 0.2}$ | $40.7_{\pm 0.4}$ | $51.0_{\pm 0.3}$ | $42.1_{\pm 0.5}$ | $52.0_{\pm 0.3}$ | $45.9_{\pm 0.3}$ | $53.5_{\pm 0.2}$ | $53.5_{\pm 0.2}$ | $58.8_{\pm 0.2}$ | $6.1_{\pm 0.2}$ |
| Cluster-Margin | $54.9_{\pm 0.5}$ | $40.4_{\pm 0.8}$ | $50.7_{\pm 0.6}$ | $41.5_{\pm 1.1}$ | $51.7_{\pm 0.5}$ | $45.5_{\pm 0.7}$ | $53.3_{\pm 0.4}$ | $53.3_{\pm 0.4}$ | $58.7_{\pm 0.3}$ | $6.3_{\pm 0.4}$ |
| DIAL | $55.4_{\pm 0.3}$ | $43.1_{\pm 0.6}$ | $51.4_{\pm 0.3}$ | $42.4_{\pm 0.6}$ | $52.5_{\pm 0.3}$ | $46.7_{\pm 0.4}$ | $54.2_{\pm 0.3}$ | $54.2_{\pm 0.3}$ | $59.2_{\pm 0.2}$ | $6.0_{\pm 0.2}$ |

### B.1.2 ALC over evaluation metrics across different number of checkpoints

| $|\mathcal{M}|$ | Avg Acc. | Worst Acc. | Avg Prec. | Worst Prec. | Avg F1 | Worst F1 | Adjusted Acc. | Balanced Acc. | AUROC | ECE |
|---|---|---|---|---|---|---|---|---|---|---|
| 2 | $55.3_{\pm 0.3}$ | $41.3_{\pm 0.5}$ | $51.2_{\pm 0.3}$ | $42.4_{\pm 0.6}$ | $52.2_{\pm 0.3}$ | $46.2_{\pm 0.4}$ | $53.7_{\pm 0.3}$ | $53.7_{\pm 0.3}$ | $59.0_{\pm 0.2}$ | $6.0_{\pm 0.2}$ |
| 5 | $55.2_{\pm 0.4}$ | $41.4_{\pm 0.8}$ | $51.1_{\pm 0.5}$ | $42.2_{\pm 0.9}$ | $52.2_{\pm 0.4}$ | $46.2_{\pm 0.6}$ | $53.7_{\pm 0.4}$ | $53.7_{\pm 0.4}$ | $58.9_{\pm 0.3}$ | $6.1_{\pm 0.4}$ |
| 10 | $55.4_{\pm 0.3}$ | $43.1_{\pm 0.6}$ | $51.4_{\pm 0.3}$ | $42.4_{\pm 0.6}$ | $52.5_{\pm 0.3}$ | $46.7_{\pm 0.4}$ | $54.2_{\pm 0.3}$ | $54.2_{\pm 0.3}$ | $59.2_{\pm 0.2}$ | $6.0_{\pm 0.2}$ |
| 20 | $55.1_{\pm 0.4}$ | $42.5_{\pm 1.2}$ | $51.0_{\pm 0.4}$ | $41.9_{\pm 0.7}$ | $52.1_{\pm 0.4}$ | $46.2_{\pm 0.6}$ | $53.9_{\pm 0.4}$ | $53.9_{\pm 0.4}$ | $59.0_{\pm 0.4}$ | $6.3_{\pm 0.3}$ |

### B.1.3 ALC over evaluation metrics across different query batch size and AL methods

| Algorithm | $k$ | Avg Acc. | Worst Acc. | Avg Prec. | Worst Prec. | Avg F1 | Worst F1 | Adjusted Acc. | Balanced Acc. | AUROC | ECE |
|---|---|---|---|---|---|---|---|---|---|---|---|
| DIAL | 25 | $69.0_{\pm 0.5}$ | $52.8_{\pm 1.1}$ | $63.9_{\pm 0.6}$ | $52.5_{\pm 0.9}$ | $65.2_{\pm 0.6}$ | $57.8_{\pm 0.8}$ | $67.3_{\pm 0.5}$ | $67.3_{\pm 0.5}$ | $73.8_{\pm 0.4}$ | $7.7_{\pm 0.4}$ |
| Margin | | $69.0_{\pm 0.4}$ | $51.3_{\pm 0.4}$ | $63.8_{\pm 0.5}$ | $52.7_{\pm 1.0}$ | $65.1_{\pm 0.5}$ | $57.6_{\pm 0.6}$ | $67.0_{\pm 0.3}$ | $67.0_{\pm 0.3}$ | $73.7_{\pm 0.2}$ | $7.7_{\pm 0.4}$ |
| DIAL | 50 | $34.6_{\pm 0.2}$ | $26.5_{\pm 0.6}$ | $32.0_{\pm 0.2}$ | $26.4_{\pm 0.4}$ | $32.7_{\pm 0.2}$ | $29.0_{\pm 0.3}$ | $33.7_{\pm 0.2}$ | $33.7_{\pm 0.2}$ | $36.9_{\pm 0.2}$ | $3.8_{\pm 0.2}$ |
| Margin | | $34.5_{\pm 0.2}$ | $25.6_{\pm 0.4}$ | $31.9_{\pm 0.3}$ | $26.4_{\pm 0.5}$ | $32.6_{\pm 0.3}$ | $28.8_{\pm 0.4}$ | $33.5_{\pm 0.2}$ | $33.5_{\pm 0.2}$ | $36.8_{\pm 0.1}$ | $3.8_{\pm 0.2}$ |
| DIAL | 100 | $17.2_{\pm 0.1}$ | $13.1_{\pm 0.4}$ | $16.0_{\pm 0.2}$ | $13.1_{\pm 0.3}$ | $16.3_{\pm 0.2}$ | $14.4_{\pm 0.2}$ | $16.8_{\pm 0.2}$ | $16.8_{\pm 0.2}$ | $18.4_{\pm 0.1}$ | $1.9_{\pm 0.1}$ |
| Margin | | $17.2_{\pm 0.1}$ | $12.9_{\pm 0.1}$ | $15.9_{\pm 0.1}$ | $13.1_{\pm 0.2}$ | $16.2_{\pm 0.1}$ | $14.3_{\pm 0.1}$ | $16.7_{\pm 0.1}$ | $16.7_{\pm 0.1}$ | $18.4_{\pm 0.1}$ | $2.0_{\pm 0.1}$ |
| DIAL | 200 | $8.6_{\pm 0.1}$ | $6.4_{\pm 0.3}$ | $7.9_{\pm 0.1}$ | $6.5_{\pm 0.1}$ | $8.1_{\pm 0.1}$ | $7.1_{\pm 0.1}$ | $8.3_{\pm 0.1}$ | $8.3_{\pm 0.1}$ | $9.2_{\pm 0.1}$ | $1.0_{\pm 0.1}$ |
| Margin | | $8.6_{\pm 0.1}$ | $6.4_{\pm 0.1}$ | $7.9_{\pm 0.1}$ | $6.5_{\pm 0.1}$ | $8.1_{\pm 0.1}$ | $7.1_{\pm 0.1}$ | $8.3_{\pm 0.0}$ | $8.3_{\pm 0.0}$ | $9.2_{\pm 0.0}$ | $1.0_{\pm 0.0}$ |

## B.2 CelebA

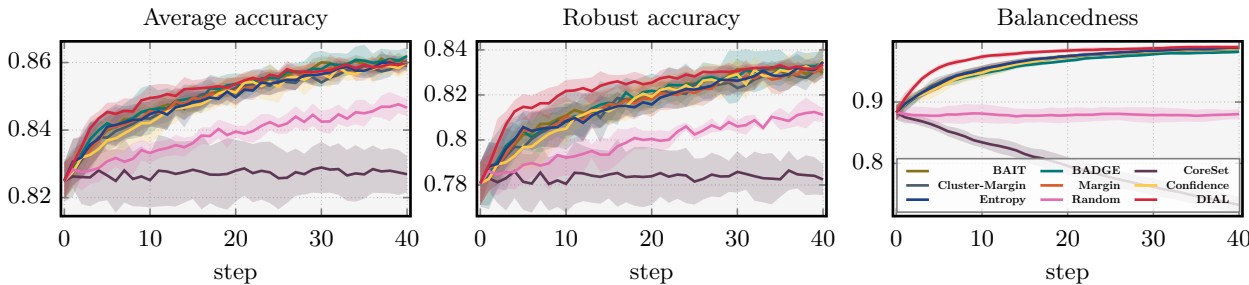

### B.2.1 The ALC over evaluation metrics across different AL methods

| Algorithm | Avg Acc. | Worst Acc. | Avg Prec. | Worst Prec. | Avg F1 | Worst F1 | Adjusted Acc. | Balanced Acc. | AUROC | ECE |
|---|---|---|---|---|---|---|---|---|---|---|
| Random | $33.5_{\pm 0.1}$ | $21.7_{\pm 1.1}$ | $32.4_{\pm 0.2}$ | $30.5_{\pm 0.7}$ | $31.1_{\pm 0.1}$ | $26.5_{\pm 0.3}$ | $30.3_{\pm 0.3}$ | $30.4_{\pm 0.3}$ | $36.0_{\pm 0.2}$ | $2.9_{\pm 0.1}$ |
| Margin | $34.0_{\pm 0.1}$ | $23.6_{\pm 1.0}$ | $32.9_{\pm 0.1}$ | $30.9_{\pm 0.2}$ | $31.9_{\pm 0.2}$ | $27.8_{\pm 0.3}$ | $31.2_{\pm 0.2}$ | $31.3_{\pm 0.2}$ | $36.5_{\pm 0.2}$ | $2.3_{\pm 0.2}$ |
| Confidence | $34.0_{\pm 0.1}$ | $23.5_{\pm 0.7}$ | $32.9_{\pm 0.1}$ | $31.0_{\pm 0.2}$ | $31.8_{\pm 0.3}$ | $27.6_{\pm 0.5}$ | $31.1_{\pm 0.4}$ | $31.1_{\pm 0.3}$ | $36.4_{\pm 0.1}$ | $2.4_{\pm 0.2}$ |
| Entropy | $34.0_{\pm 0.1}$ | $23.6_{\pm 0.9}$ | $32.9_{\pm 0.1}$ | $30.8_{\pm 0.1}$ | $31.9_{\pm 0.2}$ | $27.8_{\pm 0.3}$ | $31.2_{\pm 0.2}$ | $31.3_{\pm 0.2}$ | $36.4_{\pm 0.2}$ | $2.3_{\pm 0.2}$ |
| CoreSet | $33.1_{\pm 0.3}$ | $21.2_{\pm 1.5}$ | $31.7_{\pm 0.5}$ | $29.4_{\pm 1.0}$ | $30.6_{\pm 0.4}$ | $25.7_{\pm 0.7}$ | $29.8_{\pm 0.5}$ | $29.9_{\pm 0.5}$ | $35.5_{\pm 0.4}$ | $2.6_{\pm 0.3}$ |
| BAIT | $34.0_{\pm 0.0}$ | $23.6_{\pm 0.9}$ | $33.0_{\pm 0.1}$ | $31.1_{\pm 0.4}$ | $31.9_{\pm 0.1}$ | $27.8_{\pm 0.2}$ | $31.2_{\pm 0.2}$ | $31.3_{\pm 0.2}$ | $36.5_{\pm 0.1}$ | $2.3_{\pm 0.1}$ |
| BADGE | $34.0_{\pm 0.1}$ | $23.8_{\pm 1.9}$ | $33.0_{\pm 0.1}$ | $31.0_{\pm 0.4}$ | $32.0_{\pm 0.3}$ | $27.9_{\pm 0.6}$ | $31.3_{\pm 0.5}$ | $31.4_{\pm 0.5}$ | $36.5_{\pm 0.1}$ | $2.4_{\pm 0.3}$ |
| Cluster-Margin | $34.0_{\pm 0.1}$ | $23.5_{\pm 1.2}$ | $32.9_{\pm 0.2}$ | $31.0_{\pm 0.4}$ | $31.9_{\pm 0.2}$ | $27.7_{\pm 0.4}$ | $31.1_{\pm 0.3}$ | $31.2_{\pm 0.3}$ | $36.4_{\pm 0.1}$ | $2.3_{\pm 0.2}$ |
| DIAL | $34.1_{\pm 0.1}$ | $24.8_{\pm 1.0}$ | $33.1_{\pm 0.2}$ | $31.2_{\pm 0.5}$ | $32.0_{\pm 0.2}$ | $28.0_{\pm 0.4}$ | $31.3_{\pm 0.3}$ | $31.4_{\pm 0.3}$ | $36.6_{\pm 0.1}$ | $2.6_{\pm 0.2}$ |

### B.2.2 ALC over evaluation metrics across different number of checkpoints

| $|\mathcal{M}|$ | Avg Acc. | Worst Acc. | Avg Prec. | Worst Prec. | Avg F1 | Worst F1 | Adjusted Acc. | Balanced Acc. | AUROC | ECE |
|---|---|---|---|---|---|---|---|---|---|---|
| 2 | $34.1_{\pm 0.1}$ | $25.2_{\pm 1.4}$ | $32.9_{\pm 0.1}$ | $30.6_{\pm 0.5}$ | $32.1_{\pm 0.2}$ | $28.2_{\pm 0.5}$ | $31.5_{\pm 0.4}$ | $31.6_{\pm 0.4}$ | $36.5_{\pm 0.1}$ | $2.3_{\pm 0.2}$ |
| 4 | $34.0_{\pm 0.1}$ | $24.6_{\pm 1.6}$ | $33.0_{\pm 0.2}$ | $31.1_{\pm 0.7}$ | $32.0_{\pm 0.3}$ | $27.9_{\pm 0.5}$ | $31.2_{\pm 0.4}$ | $31.3_{\pm 0.4}$ | $36.5_{\pm 0.1}$ | $2.6_{\pm 0.3}$ |
| 5 | $34.1_{\pm 0.1}$ | $24.8_{\pm 1.0}$ | $33.1_{\pm 0.2}$ | $31.2_{\pm 0.5}$ | $32.0_{\pm 0.2}$ | $28.0_{\pm 0.4}$ | $31.3_{\pm 0.3}$ | $31.4_{\pm 0.3}$ | $36.6_{\pm 0.1}$ | $2.6_{\pm 0.2}$ |
| 10 | $34.0_{\pm 0.1}$ | $24.3_{\pm 1.0}$ | $33.0_{\pm 0.1}$ | $31.2_{\pm 0.5}$ | $31.9_{\pm 0.2}$ | $27.7_{\pm 0.4}$ | $31.1_{\pm 0.3}$ | $31.2_{\pm 0.3}$ | $36.5_{\pm 0.1}$ | $2.7_{\pm 0.2}$ |
| 20 | $34.0_{\pm 0.1}$ | $24.2_{\pm 1.4}$ | $33.0_{\pm 0.2}$ | $31.4_{\pm 0.6}$ | $31.8_{\pm 0.4}$ | $27.5_{\pm 0.7}$ | $31.0_{\pm 0.5}$ | $31.0_{\pm 0.5}$ | $36.5_{\pm 0.1}$ | $2.8_{\pm 0.3}$ |

### B.2.3 ALC over evaluation metrics across different query batch size and AL methods

| Algorithm | $k$ | Avg Acc. | Worst Acc. | Avg Prec. | Worst Prec. | Avg F1 | Worst F1 | Adjusted Acc. | Balanced Acc. | AUROC | ECE |
|---|---|---|---|---|---|---|---|---|---|---|---|
| DIAL | 10 | $85.1_{\pm 0.2}$ | $61.4_{\pm 2.8}$ | $82.5_{\pm 0.5}$ | $77.8_{\pm 1.5}$ | $79.9_{\pm 0.6}$ | $69.6_{\pm 1.1}$ | $78.1_{\pm 0.9}$ | $78.2_{\pm 0.9}$ | $91.3_{\pm 0.3}$ | $6.6_{\pm 0.5}$ |
| Margin | | $85.0_{\pm 0.3}$ | $59.2_{\pm 3.0}$ | $82.2_{\pm 0.3}$ | $77.0_{\pm 0.9}$ | $79.8_{\pm 0.6}$ | $69.6_{\pm 1.1}$ | $78.1_{\pm 0.9}$ | $78.3_{\pm 0.8}$ | $91.2_{\pm 0.3}$ | $5.8_{\pm 0.6}$ |
| DIAL | 20 | $42.6_{\pm 0.1}$ | $30.8_{\pm 1.4}$ | $41.3_{\pm 0.2}$ | $39.1_{\pm 0.7}$ | $40.0_{\pm 0.3}$ | $34.8_{\pm 0.5}$ | $39.0_{\pm 0.4}$ | $39.1_{\pm 0.4}$ | $45.7_{\pm 0.1}$ | $3.3_{\pm 0.2}$ |
| Margin | | $42.5_{\pm 0.1}$ | $29.7_{\pm 1.1}$ | $41.1_{\pm 0.3}$ | $38.6_{\pm 0.7}$ | $39.9_{\pm 0.2}$ | $34.8_{\pm 0.3}$ | $39.0_{\pm 0.2}$ | $39.1_{\pm 0.2}$ | $45.6_{\pm 0.2}$ | $2.8_{\pm 0.2}$ |
| DIAL | 25 | $34.1_{\pm 0.1}$ | $24.8_{\pm 1.0}$ | $33.1_{\pm 0.2}$ | $31.2_{\pm 0.5}$ | $32.0_{\pm 0.2}$ | $28.0_{\pm 0.4}$ | $31.3_{\pm 0.3}$ | $31.4_{\pm 0.3}$ | $36.6_{\pm 0.1}$ | $2.6_{\pm 0.2}$ |
| Margin | | $34.0_{\pm 0.1}$ | $23.6_{\pm 1.0}$ | $32.9_{\pm 0.1}$ | $30.9_{\pm 0.2}$ | $31.9_{\pm 0.2}$ | $27.8_{\pm 0.3}$ | $31.2_{\pm 0.2}$ | $31.3_{\pm 0.2}$ | $36.5_{\pm 0.2}$ | $2.3_{\pm 0.2}$ |
| DIAL | 50 | $17.0_{\pm 0.1}$ | $12.2_{\pm 0.7}$ | $16.5_{\pm 0.3}$ | $15.6_{\pm 0.3}$ | $16.0_{\pm 0.2}$ | $13.9_{\pm 0.3}$ | $15.6_{\pm 0.2}$ | $15.6_{\pm 0.2}$ | $18.3_{\pm 0.0}$ | $1.4_{\pm 0.2}$ |
| Margin | | $17.0_{\pm 0.1}$ | $11.9_{\pm 0.4}$ | $16.4_{\pm 0.1}$ | $15.4_{\pm 0.3}$ | $16.0_{\pm 0.1}$ | $13.9_{\pm 0.1}$ | $15.6_{\pm 0.1}$ | $15.7_{\pm 0.1}$ | $18.2_{\pm 0.1}$ | $1.2_{\pm 0.1}$ |
| DIAL | 100 | $8.5_{\pm 0.0}$ | $6.0_{\pm 0.4}$ | $8.3_{\pm 0.0}$ | $7.8_{\pm 0.1}$ | $8.0_{\pm 0.1}$ | $6.9_{\pm 0.2}$ | $7.8_{\pm 0.1}$ | $7.8_{\pm 0.1}$ | $9.1_{\pm 0.0}$ | $0.7_{\pm 0.1}$ |
| Margin | | $8.5_{\pm 0.0}$ | $6.0_{\pm 0.1}$ | $8.2_{\pm 0.0}$ | $7.7_{\pm 0.1}$ | $8.0_{\pm 0.0}$ | $7.0_{\pm 0.0}$ | $7.8_{\pm 0.0}$ | $7.8_{\pm 0.0}$ | $9.1_{\pm 0.0}$ | $0.6_{\pm 0.0}$ |

## B.3 Corrupted CIFAR-10

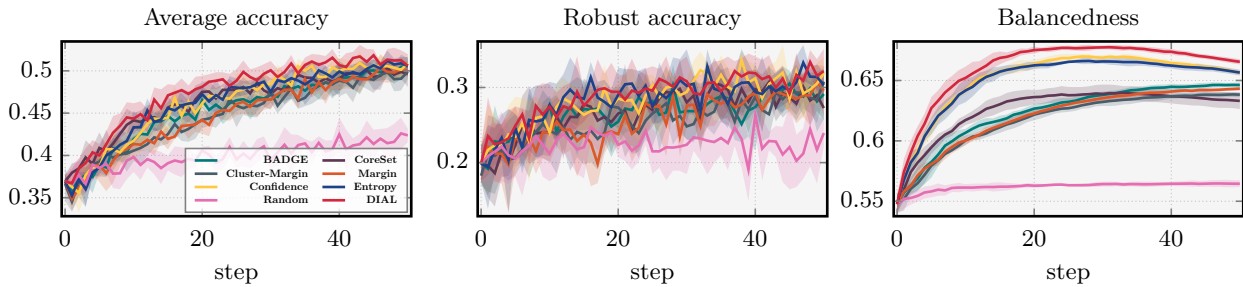

### B.3.1 ALC over evaluation metrics across different AL methods

| Algorithm | Avg Acc. | Worst Acc. | Avg Prec. | Worst Prec. | Avg F1 | Worst F1 | Adjusted Acc. | Balanced Acc. | AUROC | ECE |
|---|---|---|---|---|---|---|---|---|---|---|
| Random | $20.0_{\pm 0.4}$ | $0.2_{\pm 0.0}$ | $20.8_{\pm 0.4}$ | $11.2_{\pm 0.4}$ | $20.1_{\pm 0.4}$ | $11.4_{\pm 0.5}$ | $20.0_{\pm 0.4}$ | $20.0_{\pm 0.4}$ | $39.3_{\pm 0.2}$ | $24.7_{\pm 0.4}$ |
| Margin | $22.4_{\pm 0.2}$ | $0.6_{\pm 0.1}$ | $23.0_{\pm 0.2}$ | $12.7_{\pm 0.1}$ | $22.5_{\pm 0.2}$ | $13.1_{\pm 0.4}$ | $22.4_{\pm 0.2}$ | $22.4_{\pm 0.2}$ | $41.0_{\pm 0.1}$ | $22.4_{\pm 0.2}$ |
| Confidence | $23.1_{\pm 0.3}$ | $0.9_{\pm 0.1}$ | $23.7_{\pm 0.2}$ | $13.5_{\pm 0.2}$ | $23.2_{\pm 0.3}$ | $14.1_{\pm 0.3}$ | $23.1_{\pm 0.3}$ | $23.1_{\pm 0.3}$ | $41.6_{\pm 0.1}$ | $21.8_{\pm 0.3}$ |
| Entropy | $23.1_{\pm 0.2}$ | $0.9_{\pm 0.1}$ | $23.7_{\pm 0.2}$ | $13.5_{\pm 0.1}$ | $23.2_{\pm 0.2}$ | $14.2_{\pm 0.2}$ | $23.1_{\pm 0.2}$ | $23.1_{\pm 0.2}$ | $41.6_{\pm 0.1}$ | $21.8_{\pm 0.2}$ |
| CoreSet | $23.2_{\pm 0.3}$ | $0.7_{\pm 0.1}$ | $23.7_{\pm 0.3}$ | $13.1_{\pm 0.4}$ | $23.3_{\pm 0.3}$ | $13.6_{\pm 0.5}$ | $23.2_{\pm 0.3}$ | $23.2_{\pm 0.3}$ | $41.5_{\pm 0.2}$ | $21.3_{\pm 0.3}$ |
| BADGE | $22.7_{\pm 0.3}$ | $0.8_{\pm 0.1}$ | $23.3_{\pm 0.2}$ | $13.1_{\pm 0.2}$ | $22.8_{\pm 0.3}$ | $13.5_{\pm 0.2}$ | $22.7_{\pm 0.3}$ | $22.7_{\pm 0.3}$ | $41.3_{\pm 0.2}$ | $22.0_{\pm 0.3}$ |
| Cluster-Margin | $22.3_{\pm 0.3}$ | $0.7_{\pm 0.0}$ | $22.9_{\pm 0.3}$ | $12.9_{\pm 0.2}$ | $22.4_{\pm 0.3}$ | $13.2_{\pm 0.4}$ | $22.3_{\pm 0.3}$ | $22.3_{\pm 0.3}$ | $41.0_{\pm 0.2}$ | $22.4_{\pm 0.3}$ |
| DIAL | $23.8_{\pm 0.3}$ | $1.0_{\pm 0.2}$ | $24.3_{\pm 0.3}$ | $13.7_{\pm 0.2}$ | $23.9_{\pm 0.3}$ | $14.4_{\pm 0.3}$ | $23.8_{\pm 0.3}$ | $23.8_{\pm 0.3}$ | $42.1_{\pm 0.2}$ | $20.8_{\pm 0.3}$ |

### B.3.2 ALC over evaluation metrics across different number of checkpoints

| $|\mathcal{M}|$ | Avg Acc. | Worst Acc. | Avg Prec. | Worst Prec. | Avg F1 | Worst F1 | Adjusted Acc. | Balanced Acc. | AUROC | ECE |
|---|---|---|---|---|---|---|---|---|---|---|
| 2 | $21.8_{\pm 0.3}$ | $0.5_{\pm 0.1}$ | $22.4_{\pm 0.3}$ | $12.3_{\pm 0.3}$ | $21.9_{\pm 0.3}$ | $12.6_{\pm 0.3}$ | $21.8_{\pm 0.3}$ | $21.8_{\pm 0.3}$ | $40.6_{\pm 0.2}$ | $22.6_{\pm 0.3}$ |
| 5 | $23.4_{\pm 0.3}$ | $1.2_{\pm 0.2}$ | $23.9_{\pm 0.2}$ | $13.6_{\pm 0.2}$ | $23.5_{\pm 0.2}$ | $14.3_{\pm 0.3}$ | $23.4_{\pm 0.3}$ | $23.4_{\pm 0.3}$ | $41.8_{\pm 0.1}$ | $21.3_{\pm 0.2}$ |
| 20 | $23.6_{\pm 0.2}$ | $1.2_{\pm 0.3}$ | $24.2_{\pm 0.2}$ | $13.6_{\pm 0.3}$ | $23.7_{\pm 0.2}$ | $14.4_{\pm 0.3}$ | $23.6_{\pm 0.2}$ | $23.6_{\pm 0.2}$ | $42.0_{\pm 0.1}$ | $21.0_{\pm 0.2}$ |
| 50 | $23.8_{\pm 0.3}$ | $1.0_{\pm 0.2}$ | $24.3_{\pm 0.3}$ | $13.7_{\pm 0.2}$ | $23.9_{\pm 0.3}$ | $14.4_{\pm 0.3}$ | $23.8_{\pm 0.3}$ | $23.8_{\pm 0.3}$ | $42.1_{\pm 0.2}$ | $20.8_{\pm 0.3}$ |

### B.3.3 ALC over evaluation metrics across different query batch size and AL methods

| Algorithm | $k$ | Avg Acc. | Worst Acc. | Avg Prec. | Worst Prec. | Avg F1 | Worst F1 | Adjusted Acc. | Balanced Acc. | AUROC | ECE |
|---|---|---|---|---|---|---|---|---|---|---|---|
| Margin | 100 | $44.1_{\pm 0.4}$ | $1.1_{\pm 0.1}$ | $45.3_{\pm 0.3}$ | $25.0_{\pm 0.6}$ | $44.3_{\pm 0.4}$ | $25.9_{\pm 0.8}$ | $44.1_{\pm 0.4}$ | $44.1_{\pm 0.4}$ | $81.6_{\pm 0.3}$ | $46.0_{\pm 0.5}$ |
| DIAL | | $46.9_{\pm 0.3}$ | $1.8_{\pm 0.1}$ | $47.9_{\pm 0.3}$ | $27.0_{\pm 0.5}$ | $47.1_{\pm 0.3}$ | $28.2_{\pm 0.3}$ | $46.9_{\pm 0.3}$ | $46.9_{\pm 0.3}$ | $83.8_{\pm 0.2}$ | $43.0_{\pm 0.4}$ |
| Margin | 200 | $22.4_{\pm 0.2}$ | $0.6_{\pm 0.1}$ | $23.0_{\pm 0.2}$ | $12.7_{\pm 0.1}$ | $22.5_{\pm 0.2}$ | $13.1_{\pm 0.4}$ | $22.4_{\pm 0.2}$ | $22.4_{\pm 0.2}$ | $41.0_{\pm 0.1}$ | $22.4_{\pm 0.2}$ |
| DIAL | | $23.8_{\pm 0.3}$ | $1.0_{\pm 0.2}$ | $24.3_{\pm 0.3}$ | $13.7_{\pm 0.2}$ | $23.9_{\pm 0.3}$ | $14.4_{\pm 0.3}$ | $23.8_{\pm 0.3}$ | $23.8_{\pm 0.3}$ | $42.1_{\pm 0.2}$ | $20.8_{\pm 0.3}$ |
| Margin | 250 | $17.8_{\pm 0.2}$ | $0.6_{\pm 0.2}$ | $18.4_{\pm 0.2}$ | $10.2_{\pm 0.2}$ | $17.9_{\pm 0.2}$ | $10.6_{\pm 0.2}$ | $17.8_{\pm 0.2}$ | $17.8_{\pm 0.2}$ | $32.8_{\pm 0.1}$ | $17.8_{\pm 0.2}$ |
| DIAL | | $19.0_{\pm 0.1}$ | $0.9_{\pm 0.1}$ | $19.4_{\pm 0.1}$ | $11.1_{\pm 0.1}$ | $19.1_{\pm 0.1}$ | $11.5_{\pm 0.1}$ | $19.0_{\pm 0.1}$ | $19.0_{\pm 0.1}$ | $33.7_{\pm 0.1}$ | $16.6_{\pm 0.1}$ |
| Margin | 500 | $8.9_{\pm 0.1}$ | $0.3_{\pm 0.1}$ | $9.2_{\pm 0.1}$ | $5.2_{\pm 0.1}$ | $9.0_{\pm 0.1}$ | $5.3_{\pm 0.2}$ | $8.9_{\pm 0.1}$ | $8.9_{\pm 0.1}$ | $16.4_{\pm 0.1}$ | $8.7_{\pm 0.1}$ |
| DIAL | | $9.5_{\pm 0.1}$ | $0.4_{\pm 0.0}$ | $9.7_{\pm 0.1}$ | $5.6_{\pm 0.1}$ | $9.5_{\pm 0.1}$ | $5.7_{\pm 0.2}$ | $9.5_{\pm 0.1}$ | $9.5_{\pm 0.1}$ | $16.9_{\pm 0.0}$ | $8.1_{\pm 0.1}$ |
| Margin | 1000 | $4.5_{\pm 0.0}$ | $0.1_{\pm 0.0}$ | $4.6_{\pm 0.0}$ | $2.6_{\pm 0.1}$ | $4.5_{\pm 0.0}$ | $2.7_{\pm 0.1}$ | $4.5_{\pm 0.0}$ | $4.5_{\pm 0.0}$ | $8.2_{\pm 0.0}$ | $4.2_{\pm 0.0}$ |
| DIAL | | $4.7_{\pm 0.0}$ | $0.2_{\pm 0.0}$ | $4.8_{\pm 0.0}$ | $2.8_{\pm 0.1}$ | $4.7_{\pm 0.0}$ | $2.9_{\pm 0.1}$ | $4.7_{\pm 0.0}$ | $4.7_{\pm 0.0}$ | $8.4_{\pm 0.0}$ | $4.0_{\pm 0.0}$ |

## B.4 Some analysis

### B.4.1 DIAL upweights the minority subgroups

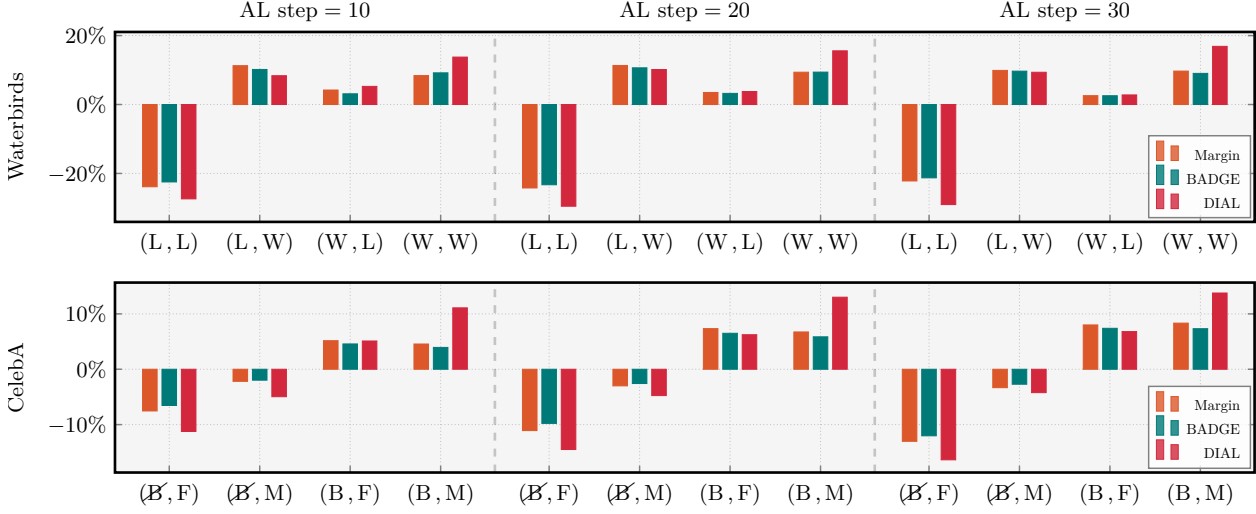

Figure B.2: The change of the distribution of $\mathcal{D}_L$ over $(Y, S)$ relative to the initial labelled pool across different AL methods. The subgroups $(\text{L}, \text{W})$ and $(\text{W}, \text{L})$ correspond *landbird in water* and *waterbird in land* respectively. Similarly, the subgroups $(\mathcal{B}, \text{F})$ and $(\text{B}, \text{M})$ correspond to *female with non-black hair* and *male with black hair*, respectively.

**Discussion.** Figure B.2 shows that AL effectively enhance the representation of underrepresented subgroups. As more samples are queried, the proportion of these underrepresented (or overrepresented) subgroups shifts: underrepresented subgroups increase while overrepresented ones decrease. Notably, DIAL demonstrates a low preference for querying the overrepresented subgroups, such as "*landbird in land*" for the Waterbirds dataset and "*non-black hair female*" for CelebA, surpassing other AL methods in this aspect.

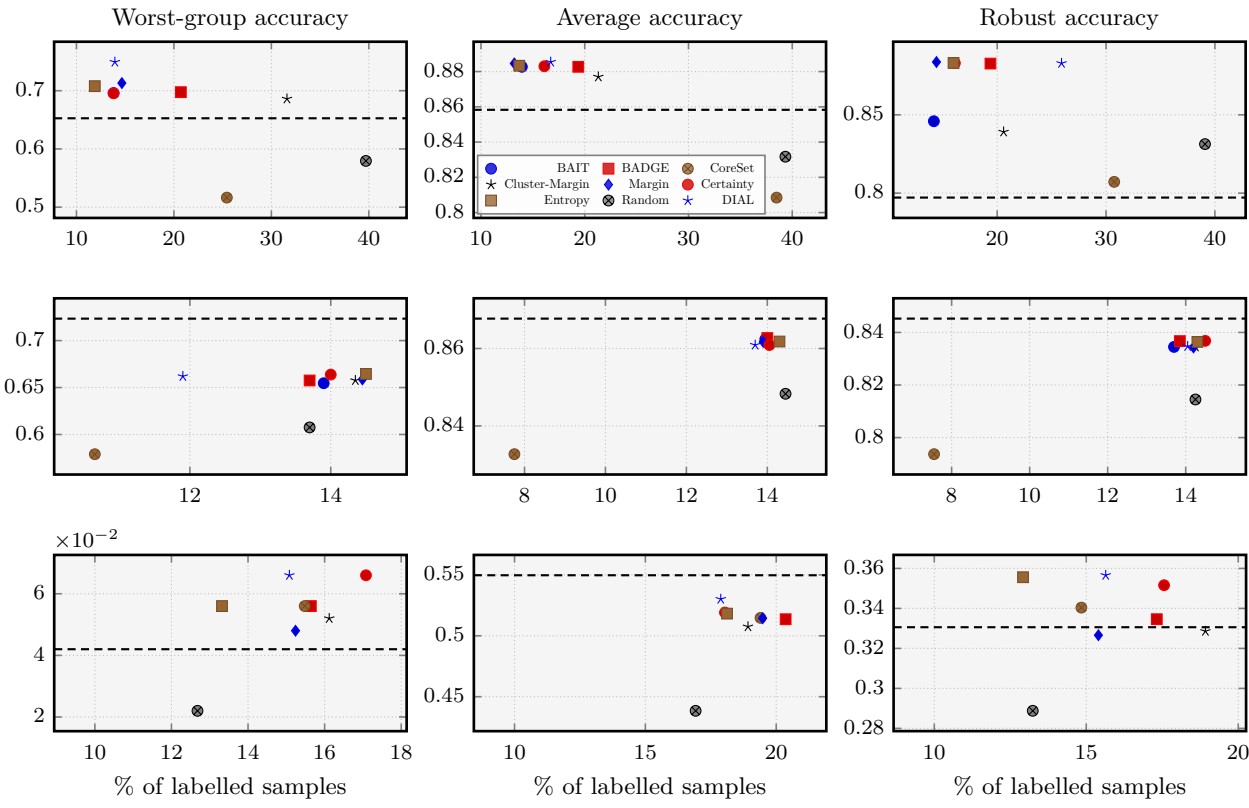

Figure B.3: **(top)** Waterbirds **(middle)** CelebA **(bottom)** Corrupted CIFAR-10. The horizontal dashed line indicates the passive performance (ERM on the compete labelled training set). The y-axis and x-axis represent the performance and the portion of labelled pool, respectively.

### B.4.2 Comparison of performance with passive setting

**Discussion.** Figure B.3 shows the performance comparison between active and passive settings. Each performance value for the AL methods is the peak achieved throughout the run. The results indicate that most AL methods consistently outperform the ERM baseline with significantly fewer samples across various performance metrics, except for Corrupted CIFAR-10. Especially, DIAL stands out as the most sample-efficient AL method among the baselines (i.e., located at the top-left corner of the plot, indicating high performance with the small number of samples), particularly for the worst-group accuracy metric.

### B.4.3 Comparison of performance with SOTA

| Dataset | Algorithm | % of samples | Avg Acc. | Worst Acc. | Avg Prec. | Worst Prec. | Avg F1 | Worst F1 | Adjusted Acc. | Balanced Acc. | AUROC | ECE |
|---|---|---|---|---|---|---|---|---|---|---|---|---|
| Waterbirds | GroupDRO | 100.0 | $86.9_{\pm 0.9}$ | $73.1_{\pm 0.4}$ | $80.7_{\pm 1.1}$ | $66.1_{\pm 2.2}$ | $82.8_{\pm 0.9}$ | $74.4_{\pm 1.2}$ | $86.3_{\pm 0.5}$ | $86.3_{\pm 0.5}$ | $94.0_{\pm 0.3}$ | $10.5_{\pm 0.8}$ |
|  | JTT | 100.0 | $88.9_{\pm 0.6}$ | $71.2_{\pm 0.5}$ | $83.2_{\pm 0.8}$ | $71.4_{\pm 1.6}$ | $84.7_{\pm 0.6}$ | $76.8_{\pm 0.8}$ | $86.8_{\pm 0.2}$ | $86.8_{\pm 0.2}$ | $94.2_{\pm 0.1}$ | $9.2_{\pm 0.3}$ |
|  | LfF | 100.0 | $86.6_{\pm 0.5}$ | $75.0_{\pm 0.7}$ | $80.3_{\pm 0.6}$ | $65.1_{\pm 1.1}$ | $82.5_{\pm 0.5}$ | $74.0_{\pm 0.7}$ | $86.3_{\pm 0.3}$ | $86.3_{\pm 0.3}$ | $93.4_{\pm 0.2}$ | $10.0_{\pm 0.8}$ |
|  | DIAL | 12.8 | $87.4_{\pm 0.6}$ | $76.4_{\pm 1.1}$ | $81.2_{\pm 0.8}$ | $66.5_{\pm 1.1}$ | $83.5_{\pm 0.8}$ | $75.4_{\pm 1.2}$ | $87.3_{\pm 0.9}$ | $87.3_{\pm 0.9}$ | $94.2_{\pm 0.7}$ | $10.1_{\pm 0.6}$ |
| CelebA | GroupDRO | 100.0 | $88.5_{\pm 0.4}$ | $82.8_{\pm 2.7}$ | $85.0_{\pm 0.8}$ | $76.2_{\pm 2.5}$ | $85.9_{\pm 0.3}$ | $79.8_{\pm 0.3}$ | $86.8_{\pm 0.6}$ | $87.1_{\pm 0.7}$ | $95.0_{\pm 0.2}$ | $4.0_{\pm 1.1}$ |
|  | JTT | 100.0 | $88.7_{\pm 0.4}$ | $73.8_{\pm 6.5}$ | $86.7_{\pm 0.9}$ | $82.4_{\pm 3.7}$ | $85.3_{\pm 1.1}$ | $78.1_{\pm 2.1}$ | $84.2_{\pm 2.3}$ | $84.4_{\pm 2.3}$ | $95.2_{\pm 0.1}$ | $4.7_{\pm 1.3}$ |
|  | LfF | 100.0 | $80.0_{\pm 1.1}$ | $64.8_{\pm 7.1}$ | $75.5_{\pm 1.3}$ | $61.7_{\pm 3.3}$ | $75.9_{\pm 1.5}$ | $66.1_{\pm 3.2}$ | $77.3_{\pm 3.0}$ | $77.7_{\pm 3.2}$ | $87.6_{\pm 1.4}$ | $6.8_{\pm 2.1}$ |
|  | DIAL | 13.5 | $87.0_{\pm 0.2}$ | $70.6_{\pm 2.8}$ | $84.3_{\pm 0.5}$ | $79.0_{\pm 1.6}$ | $83.1_{\pm 0.4}$ | $74.9_{\pm 0.8}$ | $82.0_{\pm 0.9}$ | $82.1_{\pm 0.9}$ | $92.9_{\pm 0.1}$ | $9.2_{\pm 0.3}$ |
| Corrupted CIFAR-10 | GroupDRO | 100.0 | $23.3_{\pm 1.2}$ | $23.3_{\pm 1.9}$ | – | $23.3_{\pm 1.9}$ | – | $20.6_{\pm 1.5}$ | – | $23.0_{\pm 1.1}$ | $23.2_{\pm 1.2}$ | $68.0_{\pm 2.2}$ | $7.4_{\pm 0.8}$ |
|  | JTT | 100.0 | $23.8_{\pm 1.1}$ | $20.3_{\pm 0.8}$ | – | $20.3_{\pm 0.8}$ | – | $18.2_{\pm 0.9}$ | – | $23.5_{\pm 1.0}$ | $23.6_{\pm 1.0}$ | $71.1_{\pm 1.1}$ | $4.0_{\pm 0.7}$ |
|  | LfF | 100.0 | $21.4_{\pm 1.4}$ | – | $22.2_{\pm 2.9}$ | – | $17.2_{\pm 1.8}$ | – | $21.2_{\pm 1.3}$ | $21.4_{\pm 1.3}$ | $67.8_{\pm 3.6}$ | $7.5_{\pm 6.4}$ |
|  | DIAL | 16.6 | $56.4_{\pm 0.9}$ | – | $56.8_{\pm 0.7}$ | – | $56.4_{\pm 0.8}$ | – | $56.1_{\pm 0.9}$ | $56.4_{\pm 0.9}$ | $90.0_{\pm 0.6}$ | $30.5_{\pm 1.1}$ |

Table B.2: Performance summary of DIAL and SOTA algorithms.[7]

---

[7]The values of SOTA on Waterbirds are directly taken from Yang et al. (2023), with subscripts denoting the standard error.

**Discussion.** Table B.2 presents the performance comparison between DIAL and SOTA methods: JTT (Liu et al., 2021), GroupDRO (Sagawa et al., 2020) and LfF (Nam et al., 2020). For a fair comparison, we followed the same protocol as Yang et al. (2023), performing random hyperparameter searches with 16 trials and reporting the best performance (based on the validation set without spurious attributes) over 5 trials with different random seeds. The performance of DIAL is reported at the selected step (based on the validation set), while SOTA methods report results over the full training set.

DIAL outperforms all SOTA baselines on the Waterbirds and Corrupted CIFAR-10 datasets, achieving the highest worst-group accuracy and highest average accuracy, respectively, with fewer samples. However, on CelebA, GroupDRO demonstrates superior performance, particularly in terms of worst-group accuracy. This is because GroupDRO, a robust optimization framework, explicitly minimises the worst-group expected accuracy. In the absence of spurious attribute annotations, the worst-group defaults to the worst class-wise scenario, i.e., the lowest recall across all classes. Due to class imbalance in the CelebA dataset (as shown in Figure A.1), where the two minority subgroups fall into the minority class, GroupDRO effectively improves robustness for these minority subgroups by optimising for the worst-class scenario.

Nonetheless, on CelebA, DIAL still outperforms LfF and performs close to JTT in terms of worst-group accuracy, despite using far fewer samples. For average accuracy, DIAL achieves 87%, which is comparable to JTT (88.7%) and GroupDRO (88.5%). This comparison highlights that DIAL is a competitive AL method that can achieve robustness performance under subpopulation shifts, and in some cases, outperforms the SOTA methods.

## C   Early stopping active learning

The stopping criteria in AL is used to terminate the AL loops, balancing the performance and labeling cost. Since this is not main scope in this work, we only present the early stopped results retrospectively based on some metrics proposed by Yang et al. (2023): (1) test worst-group accuracy (C.1): oracle criteria based on the *worst-group accuracy* on the test split. (2) validation worst-group accuracy (C.2): criteria based on the *worst-group accuracy* on the validation split, assuming the spurious attributes are known. (3) validation average accuracy (C.3): criteria based on the *average accuracy* on the validation split, assuming the spurious attributes are unknown.

### C.1   Test worst-group accuracy

### C.1.1   Waterbirds

| Algorithm | % of samples | Avg Acc. | Worst Acc. | Avg Prec. | Worst Prec. | Avg F1 | Worst F1 | Adjusted Acc. | Balanced Acc. | AUROC | ECE |
|---|---|---|---|---|---|---|---|---|---|---|---|
| Random | 42.5 | $82.4_{\pm1.3}$ | $55.4_{\pm4.4}$ | $75.1_{\pm1.5}$ | $58.1_{\pm2.5}$ | $76.8_{\pm1.5}$ | $65.3_{\pm2.2}$ | $79.6_{\pm1.5}$ | $79.6_{\pm1.5}$ | $89.0_{\pm1.1}$ | $11.9_{\pm1.1}$ |
| Margin | 10.4 | $86.9_{\pm1.1}$ | $69.5_{\pm2.9}$ | $80.6_{\pm1.5}$ | $66.8_{\pm2.8}$ | $82.4_{\pm1.2}$ | $73.6_{\pm1.7}$ | $85.1_{\pm0.9}$ | $85.1_{\pm0.9}$ | $92.8_{\pm0.5}$ | $8.6_{\pm0.9}$ |
| Confidence | 10.4 | $87.1_{\pm0.6}$ | $67.1_{\pm3.4}$ | $80.9_{\pm0.8}$ | $67.6_{\pm1.9}$ | $82.5_{\pm0.7}$ | $73.5_{\pm0.9}$ | $84.7_{\pm0.7}$ | $84.7_{\pm0.7}$ | $92.6_{\pm0.3}$ | $8.2_{\pm0.6}$ |
| Entropy | 10.4 | $86.8_{\pm1.5}$ | $68.2_{\pm3.3}$ | $80.5_{\pm1.9}$ | $66.8_{\pm3.8}$ | $82.2_{\pm1.6}$ | $73.2_{\pm2.3}$ | $84.8_{\pm1.1}$ | $84.8_{\pm1.1}$ | $92.6_{\pm0.7}$ | $8.5_{\pm1.4}$ |
| BAIT | 13.5 | $87.1_{\pm0.7}$ | $68.0_{\pm1.2}$ | $80.9_{\pm1.0}$ | $67.4_{\pm1.8}$ | $82.6_{\pm0.9}$ | $73.7_{\pm1.2}$ | $85.1_{\pm0.6}$ | $85.1_{\pm0.6}$ | $92.8_{\pm0.4}$ | $8.7_{\pm0.6}$ |
| CoreSet | 36.4 | $79.3_{\pm5.4}$ | $47.5_{\pm14.4}$ | $71.7_{\pm5.9}$ | $53.0_{\pm9.2}$ | $73.1_{\pm6.3}$ | $60.2_{\pm8.8}$ | $76.0_{\pm6.2}$ | $76.0_{\pm6.2}$ | $85.7_{\pm5.1}$ | $13.9_{\pm3.4}$ |
| Cluster-Margin | 21.9 | $86.6_{\pm0.6}$ | $66.2_{\pm0.6}$ | $80.2_{\pm0.8}$ | $66.2_{\pm1.5}$ | $81.9_{\pm0.7}$ | $72.7_{\pm0.9}$ | $84.4_{\pm0.4}$ | $84.4_{\pm0.4}$ | $92.5_{\pm0.2}$ | $9.5_{\pm0.5}$ |
| BADGE | 14.1 | $87.3_{\pm0.5}$ | $68.2_{\pm1.1}$ | $81.1_{\pm0.7}$ | $67.8_{\pm1.5}$ | $82.8_{\pm0.5}$ | $74.0_{\pm0.7}$ | $85.2_{\pm0.2}$ | $85.2_{\pm0.2}$ | $92.9_{\pm0.2}$ | $8.6_{\pm0.6}$ |
| DIAL | 12.8 | $87.0_{\pm1.0}$ | $74.0_{\pm1.7}$ | $80.7_{\pm1.1}$ | $66.0_{\pm2.3}$ | $82.8_{\pm1.0}$ | $74.4_{\pm1.3}$ | $86.4_{\pm0.5}$ | $86.4_{\pm0.5}$ | $93.5_{\pm0.4}$ | $9.4_{\pm1.0}$ |

### C.1.2   CelebA

| Algorithm | % of samples | Avg Acc. | Worst Acc. | Avg Prec. | Worst Prec. | Avg F1 | Worst F1 | Adjusted Acc. | Balanced Acc. | AUROC | ECE |
|---|---|---|---|---|---|---|---|---|---|---|---|
| Random | 14.5 | $84.7_{\pm0.4}$ | $59.5_{\pm3.1}$ | $81.7_{\pm0.4}$ | $76.3_{\pm1.2}$ | $79.5_{\pm1.0}$ | $69.2_{\pm1.8}$ | $77.8_{\pm1.4}$ | $78.0_{\pm1.3}$ | $91.0_{\pm0.4}$ | $6.8_{\pm0.9}$ |
| Margin | 15.0 | $86.0_{\pm0.2}$ | $64.3_{\pm3.0}$ | $83.3_{\pm0.3}$ | $78.4_{\pm1.2}$ | $81.4_{\pm0.5}$ | $72.1_{\pm0.9}$ | $79.8_{\pm0.8}$ | $80.0_{\pm0.8}$ | $92.2_{\pm0.2}$ | $4.7_{\pm1.0}$ |
| Confidence | 15.0 | $86.0_{\pm0.3}$ | $65.3_{\pm3.1}$ | $83.2_{\pm0.4}$ | $78.0_{\pm0.7}$ | $81.5_{\pm0.5}$ | $72.4_{\pm0.8}$ | $80.1_{\pm0.6}$ | $80.2_{\pm0.6}$ | $92.2_{\pm0.3}$ | $5.0_{\pm0.7}$ |
| Entropy | 15.0 | $86.0_{\pm0.2}$ | $65.0_{\pm3.5}$ | $83.2_{\pm0.2}$ | $78.0_{\pm1.3}$ | $81.5_{\pm0.6}$ | $72.4_{\pm1.1}$ | $80.1_{\pm1.0}$ | $80.2_{\pm1.0}$ | $92.2_{\pm0.1}$ | $4.8_{\pm0.9}$ |
| BAIT | 12.8 | $86.0_{\pm0.2}$ | $64.1_{\pm2.1}$ | $83.3_{\pm0.3}$ | $78.1_{\pm0.9}$ | $81.5_{\pm0.4}$ | $72.3_{\pm0.8}$ | $80.0_{\pm0.7}$ | $80.1_{\pm0.7}$ | $92.3_{\pm0.1}$ | $5.0_{\pm0.5}$ |
| CoreSet | 10.8 | $82.8_{\pm0.9}$ | $55.2_{\pm4.3}$ | $79.1_{\pm1.2}$ | $72.4_{\pm2.1}$ | $76.9_{\pm1.3}$ | $65.2_{\pm2.2}$ | $75.1_{\pm1.5}$ | $75.4_{\pm1.5}$ | $88.8_{\pm1.0}$ | $5.9_{\pm1.0}$ |
| Cluster-Margin | 15.0 | $85.9_{\pm0.2}$ | $64.6_{\pm2.2}$ | $83.1_{\pm0.4}$ | $77.9_{\pm1.3}$ | $81.4_{\pm0.4}$ | $72.2_{\pm0.7}$ | $79.9_{\pm0.7}$ | $80.1_{\pm0.7}$ | $92.1_{\pm0.3}$ | $5.1_{\pm0.6}$ |
| BADGE | 15.0 | $86.2_{\pm0.3}$ | $64.1_{\pm3.2}$ | $83.7_{\pm0.2}$ | $79.0_{\pm0.7}$ | $81.6_{\pm0.6}$ | $72.5_{\pm1.0}$ | $80.1_{\pm0.8}$ | $80.2_{\pm0.8}$ | $92.4_{\pm0.2}$ | $5.4_{\pm0.5}$ |
| DIAL | 11.0 | $85.6_{\pm0.2}$ | $64.5_{\pm2.3}$ | $82.9_{\pm0.2}$ | $78.0_{\pm0.5}$ | $80.9_{\pm0.3}$ | $71.3_{\pm0.6}$ | $79.2_{\pm0.5}$ | $79.4_{\pm0.5}$ | $91.8_{\pm0.1}$ | $6.2_{\pm0.2}$ |

## C.2 Validation worst-group accuracy

### C.2.1 Waterbirds

| Algorithm | % of samples | Avg Acc. | Worst Acc. | Avg Prec. | Worst Prec. | Avg F1 | Worst F1 | Adjusted Acc. | Balanced Acc. | AUROC | ECE |
|---|---|---|---|---|---|---|---|---|---|---|---|
| Random | 41.9 | $82.0_{\pm 1.4}$ | $55.2_{\pm 2.9}$ | $74.7_{\pm 1.5}$ | $57.2_{\pm 2.6}$ | $76.3_{\pm 1.5}$ | $64.8_{\pm 2.0}$ | $79.4_{\pm 1.3}$ | $79.4_{\pm 1.3}$ | $88.8_{\pm 1.1}$ | $12.3_{\pm 1.3}$ |
| Margin | 10.4 | $86.9_{\pm 1.1}$ | $69.5_{\pm 2.9}$ | $80.6_{\pm 1.5}$ | $66.8_{\pm 2.8}$ | $82.4_{\pm 1.2}$ | $73.6_{\pm 1.7}$ | $85.1_{\pm 0.9}$ | $85.1_{\pm 0.9}$ | $92.8_{\pm 0.5}$ | $8.6_{\pm 0.9}$ |
| Confidence | 10.4 | $87.1_{\pm 0.6}$ | $67.1_{\pm 3.4}$ | $80.9_{\pm 0.8}$ | $67.6_{\pm 1.9}$ | $82.5_{\pm 0.7}$ | $73.5_{\pm 0.9}$ | $84.7_{\pm 0.7}$ | $84.7_{\pm 0.7}$ | $92.6_{\pm 0.3}$ | $8.2_{\pm 0.6}$ |
| Entropy | 10.4 | $86.8_{\pm 1.5}$ | $68.2_{\pm 3.3}$ | $80.5_{\pm 1.9}$ | $66.8_{\pm 3.8}$ | $82.2_{\pm 1.6}$ | $73.2_{\pm 2.3}$ | $84.8_{\pm 1.1}$ | $84.8_{\pm 1.1}$ | $92.6_{\pm 0.7}$ | $8.5_{\pm 1.4}$ |
| BAIT | 13.5 | $87.1_{\pm 0.7}$ | $68.0_{\pm 1.2}$ | $80.9_{\pm 1.0}$ | $67.4_{\pm 1.8}$ | $82.6_{\pm 0.9}$ | $73.7_{\pm 1.2}$ | $85.1_{\pm 0.6}$ | $85.1_{\pm 0.6}$ | $92.8_{\pm 0.4}$ | $8.7_{\pm 0.6}$ |
| CoreSet | 36.4 | $79.3_{\pm 5.4}$ | $47.5_{\pm 14.4}$ | $71.7_{\pm 5.9}$ | $53.0_{\pm 9.2}$ | $73.1_{\pm 6.3}$ | $60.2_{\pm 8.8}$ | $76.0_{\pm 6.2}$ | $76.0_{\pm 6.2}$ | $85.7_{\pm 5.1}$ | $13.9_{\pm 3.4}$ |
| Cluster-Margin | 21.3 | $86.7_{\pm 0.7}$ | $66.0_{\pm 1.6}$ | $80.4_{\pm 0.9}$ | $66.7_{\pm 1.7}$ | $82.0_{\pm 0.8}$ | $72.9_{\pm 1.1}$ | $84.4_{\pm 0.6}$ | $84.4_{\pm 0.6}$ | $92.5_{\pm 0.3}$ | $9.4_{\pm 0.5}$ |
| BADGE | 14.1 | $87.3_{\pm 0.5}$ | $68.2_{\pm 1.1}$ | $81.1_{\pm 0.7}$ | $67.8_{\pm 1.5}$ | $82.8_{\pm 0.5}$ | $74.0_{\pm 0.7}$ | $85.2_{\pm 0.2}$ | $85.2_{\pm 0.2}$ | $92.9_{\pm 0.2}$ | $8.6_{\pm 0.6}$ |
| DIAL | 12.8 | $87.0_{\pm 1.0}$ | $74.0_{\pm 1.7}$ | $80.7_{\pm 1.1}$ | $66.0_{\pm 2.3}$ | $82.8_{\pm 1.0}$ | $74.4_{\pm 1.3}$ | $86.4_{\pm 0.5}$ | $86.4_{\pm 0.5}$ | $93.5_{\pm 0.4}$ | $9.4_{\pm 1.0}$ |

### C.2.2 CelebA

| Algorithm | % of samples | Avg Acc. | Worst Acc. | Avg Prec. | Worst Prec. | Avg F1 | Worst F1 | Adjusted Acc. | Balanced Acc. | AUROC | ECE |
|---|---|---|---|---|---|---|---|---|---|---|---|
| Random | 14.5 | $84.7_{\pm 0.4}$ | $59.5_{\pm 3.1}$ | $81.7_{\pm 0.4}$ | $76.3_{\pm 1.2}$ | $79.5_{\pm 1.0}$ | $69.2_{\pm 1.8}$ | $77.8_{\pm 1.4}$ | $78.0_{\pm 1.3}$ | $91.0_{\pm 0.4}$ | $6.8_{\pm 0.9}$ |
| Margin | 14.5 | $86.1_{\pm 0.3}$ | $64.1_{\pm 2.5}$ | $83.4_{\pm 0.2}$ | $78.3_{\pm 0.6}$ | $81.6_{\pm 0.5}$ | $72.4_{\pm 1.0}$ | $80.1_{\pm 0.8}$ | $80.2_{\pm 0.8}$ | $92.2_{\pm 0.2}$ | $4.8_{\pm 0.5}$ |
| Confidence | 15.0 | $86.0_{\pm 0.3}$ | $65.3_{\pm 3.1}$ | $83.2_{\pm 0.4}$ | $78.0_{\pm 0.7}$ | $81.5_{\pm 0.5}$ | $72.4_{\pm 0.8}$ | $80.1_{\pm 0.6}$ | $80.2_{\pm 0.6}$ | $92.2_{\pm 0.3}$ | $5.0_{\pm 0.7}$ |
| Entropy | 13.3 | $85.9_{\pm 0.1}$ | $64.4_{\pm 1.1}$ | $82.9_{\pm 0.3}$ | $77.3_{\pm 0.6}$ | $81.4_{\pm 0.2}$ | $72.3_{\pm 0.3}$ | $80.1_{\pm 0.2}$ | $80.3_{\pm 0.2}$ | $92.0_{\pm 0.2}$ | $4.9_{\pm 0.5}$ |
| BAIT | 14.8 | $86.1_{\pm 0.2}$ | $64.1_{\pm 2.3}$ | $83.4_{\pm 0.4}$ | $78.3_{\pm 1.4}$ | $81.6_{\pm 0.5}$ | $72.5_{\pm 1.0}$ | $80.1_{\pm 0.9}$ | $80.3_{\pm 0.9}$ | $92.4_{\pm 0.1}$ | $4.9_{\pm 0.6}$ |
| CoreSet | 10.8 | $82.8_{\pm 0.9}$ | $55.2_{\pm 4.3}$ | $79.1_{\pm 1.2}$ | $72.4_{\pm 2.1}$ | $76.9_{\pm 1.3}$ | $65.2_{\pm 2.2}$ | $75.1_{\pm 1.5}$ | $75.4_{\pm 1.5}$ | $88.8_{\pm 1.0}$ | $5.9_{\pm 1.0}$ |
| Cluster-Margin | 15.0 | $85.9_{\pm 0.2}$ | $64.6_{\pm 2.2}$ | $83.1_{\pm 0.4}$ | $77.9_{\pm 1.3}$ | $81.4_{\pm 0.4}$ | $72.2_{\pm 0.7}$ | $79.9_{\pm 0.7}$ | $80.1_{\pm 0.7}$ | $92.1_{\pm 0.3}$ | $5.1_{\pm 0.6}$ |
| BADGE | 14.5 | $86.1_{\pm 0.3}$ | $63.8_{\pm 3.4}$ | $83.5_{\pm 0.2}$ | $78.6_{\pm 0.5}$ | $81.5_{\pm 0.6}$ | $72.4_{\pm 1.0}$ | $80.0_{\pm 0.8}$ | $80.1_{\pm 0.8}$ | $92.4_{\pm 0.1}$ | $5.4_{\pm 0.6}$ |
| DIAL | 11.0 | $85.6_{\pm 0.2}$ | $64.5_{\pm 2.3}$ | $82.9_{\pm 0.2}$ | $78.0_{\pm 0.5}$ | $80.9_{\pm 0.3}$ | $71.3_{\pm 0.6}$ | $79.2_{\pm 0.5}$ | $79.4_{\pm 0.5}$ | $91.8_{\pm 0.1}$ | $6.2_{\pm 0.2}$ |

## C.3 Validation average accuracy

### C.3.1 Waterbirds

| Algorithm | % of samples | Avg Acc. | Worst Acc. | Avg Prec. | Worst Prec. | Avg F1 | Worst F1 | Adjusted Acc. | Balanced Acc. | AUROC | ECE |
|---|---|---|---|---|---|---|---|---|---|---|---|
| Random | 42.5 | $82.4_{\pm 1.3}$ | $55.4_{\pm 4.4}$ | $75.1_{\pm 1.5}$ | $58.1_{\pm 2.5}$ | $76.8_{\pm 1.5}$ | $65.3_{\pm 2.2}$ | $79.6_{\pm 1.5}$ | $79.6_{\pm 1.5}$ | $89.0_{\pm 1.1}$ | $11.9_{\pm 1.1}$ |
| Margin | 17.7 | $87.9_{\pm 0.7}$ | $65.1_{\pm 2.8}$ | $82.0_{\pm 1.0}$ | $70.0_{\pm 2.2}$ | $83.3_{\pm 0.8}$ | $74.5_{\pm 1.0}$ | $85.0_{\pm 0.6}$ | $85.0_{\pm 0.6}$ | $93.0_{\pm 0.3}$ | $8.3_{\pm 0.7}$ |
| Confidence | 13.5 | $87.6_{\pm 0.8}$ | $65.4_{\pm 2.3}$ | $81.6_{\pm 1.0}$ | $69.1_{\pm 2.0}$ | $83.0_{\pm 0.9}$ | $74.1_{\pm 1.2}$ | $84.9_{\pm 0.7}$ | $84.9_{\pm 0.7}$ | $92.9_{\pm 0.3}$ | $7.9_{\pm 0.7}$ |
| Entropy | 11.0 | $87.6_{\pm 1.0}$ | $67.1_{\pm 4.4}$ | $81.5_{\pm 1.3}$ | $68.7_{\pm 2.0}$ | $83.0_{\pm 1.3}$ | $74.2_{\pm 1.9}$ | $85.1_{\pm 1.4}$ | $85.1_{\pm 1.4}$ | $92.8_{\pm 0.6}$ | $7.8_{\pm 0.6}$ |
| BAIT | 11.6 | $87.7_{\pm 0.9}$ | $65.5_{\pm 4.1}$ | $81.7_{\pm 1.2}$ | $69.4_{\pm 2.5}$ | $83.0_{\pm 1.1}$ | $74.1_{\pm 1.5}$ | $84.8_{\pm 1.0}$ | $84.8_{\pm 1.0}$ | $92.6_{\pm 0.5}$ | $7.9_{\pm 0.8}$ |
| CoreSet | 38.9 | $80.0_{\pm 5.2}$ | $44.7_{\pm 13.8}$ | $72.3_{\pm 5.9}$ | $54.4_{\pm 9.3}$ | $73.6_{\pm 6.2}$ | $60.6_{\pm 8.7}$ | $75.9_{\pm 6.0}$ | $75.9_{\pm 6.0}$ | $85.9_{\pm 4.8}$ | $13.2_{\pm 3.5}$ |
| Cluster-Margin | 24.9 | $87.0_{\pm 0.6}$ | $64.7_{\pm 2.3}$ | $80.7_{\pm 0.8}$ | $67.6_{\pm 1.5}$ | $82.3_{\pm 0.7}$ | $73.1_{\pm 0.9}$ | $84.4_{\pm 0.6}$ | $84.4_{\pm 0.6}$ | $92.6_{\pm 0.3}$ | $9.1_{\pm 0.5}$ |
| BADGE | 13.5 | $87.6_{\pm 0.3}$ | $66.4_{\pm 1.2}$ | $81.5_{\pm 0.4}$ | $69.0_{\pm 0.8}$ | $83.0_{\pm 0.3}$ | $74.2_{\pm 0.4}$ | $85.0_{\pm 0.2}$ | $85.0_{\pm 0.2}$ | $92.9_{\pm 0.2}$ | $8.1_{\pm 0.3}$ |
| DIAL | 20.7 | $88.1_{\pm 0.5}$ | $69.2_{\pm 1.5}$ | $82.2_{\pm 0.8}$ | $69.7_{\pm 1.7}$ | $83.7_{\pm 0.6}$ | $75.3_{\pm 0.7}$ | $85.9_{\pm 0.3}$ | $85.9_{\pm 0.3}$ | $93.5_{\pm 0.2}$ | $8.2_{\pm 0.5}$ |

### C.3.2 CelebA

| Algorithm | % of samples | Avg Acc. | Worst Acc. | Avg Prec. | Worst Prec. | Avg F1 | Worst F1 | Adjusted Acc. | Balanced Acc. | AUROC | ECE |
|---|---|---|---|---|---|---|---|---|---|---|---|
| Random | 14.8 | $84.8_{\pm 0.3}$ | $59.3_{\pm 1.1}$ | $81.8_{\pm 0.5}$ | $76.4_{\pm 1.1}$ | $79.6_{\pm 0.4}$ | $69.4_{\pm 0.6}$ | $77.9_{\pm 0.5}$ | $78.1_{\pm 0.5}$ | $91.0_{\pm 0.3}$ | $6.6_{\pm 0.5}$ |
| Margin | 14.5 | $86.1_{\pm 0.3}$ | $64.1_{\pm 2.5}$ | $83.4_{\pm 0.2}$ | $78.3_{\pm 0.6}$ | $81.6_{\pm 0.5}$ | $72.4_{\pm 1.0}$ | $80.1_{\pm 0.8}$ | $80.2_{\pm 0.8}$ | $92.2_{\pm 0.2}$ | $4.8_{\pm 0.5}$ |
| Confidence | 14.2 | $85.9_{\pm 0.3}$ | $63.9_{\pm 2.5}$ | $83.3_{\pm 0.2}$ | $78.4_{\pm 0.6}$ | $81.2_{\pm 0.6}$ | $71.9_{\pm 1.1}$ | $79.7_{\pm 0.9}$ | $79.8_{\pm 0.9}$ | $92.1_{\pm 0.3}$ | $5.3_{\pm 0.5}$ |
| Entropy | 14.5 | $86.0_{\pm 0.2}$ | $63.9_{\pm 2.2}$ | $83.3_{\pm 0.2}$ | $78.1_{\pm 0.7}$ | $81.5_{\pm 0.4}$ | $72.3_{\pm 0.8}$ | $80.0_{\pm 0.6}$ | $80.2_{\pm 0.6}$ | $92.1_{\pm 0.1}$ | $4.7_{\pm 0.5}$ |
| BAIT | 14.2 | $86.1_{\pm 0.2}$ | $63.6_{\pm 2.2}$ | $83.5_{\pm 0.2}$ | $78.7_{\pm 0.7}$ | $81.6_{\pm 0.5}$ | $72.4_{\pm 0.9}$ | $80.0_{\pm 0.7}$ | $80.1_{\pm 0.7}$ | $92.3_{\pm 0.2}$ | $5.0_{\pm 0.6}$ |
| CoreSet | 7.8 | $82.8_{\pm 1.1}$ | $52.2_{\pm 3.3}$ | $79.6_{\pm 1.7}$ | $74.2_{\pm 3.0}$ | $76.4_{\pm 1.4}$ | $64.0_{\pm 2.1}$ | $74.3_{\pm 1.4}$ | $74.5_{\pm 1.3}$ | $88.8_{\pm 1.4}$ | $6.5_{\pm 0.9}$ |
| Cluster-Margin | 14.8 | $86.0_{\pm 0.1}$ | $64.1_{\pm 0.9}$ | $83.2_{\pm 0.3}$ | $78.1_{\pm 0.9}$ | $81.4_{\pm 0.1}$ | $72.2_{\pm 0.3}$ | $79.9_{\pm 0.3}$ | $80.0_{\pm 0.3}$ | $92.2_{\pm 0.3}$ | $5.0_{\pm 0.5}$ |
| BADGE | 14.5 | $86.1_{\pm 0.3}$ | $63.8_{\pm 3.4}$ | $83.5_{\pm 0.2}$ | $78.6_{\pm 0.5}$ | $81.5_{\pm 0.6}$ | $72.4_{\pm 1.0}$ | $80.0_{\pm 0.8}$ | $80.1_{\pm 0.8}$ | $92.4_{\pm 0.1}$ | $5.4_{\pm 0.6}$ |
| DIAL | 14.8 | $85.9_{\pm 0.2}$ | $63.7_{\pm 1.8}$ | $83.6_{\pm 0.2}$ | $79.4_{\pm 0.6}$ | $81.1_{\pm 0.3}$ | $71.5_{\pm 0.6}$ | $79.2_{\pm 0.5}$ | $79.4_{\pm 0.5}$ | $92.2_{\pm 0.1}$ | $6.5_{\pm 0.5}$ |

