# OpenReview forum: "Assessing and enhancing robustness of active learning strategies to spurious bias"
_TMLR — Rejected by TMLR_

### Review · Reviewer_bAF9 · 2024-08-21

**Summary Of Contributions:**

The paper introduces Domain-Invariant Active Learning (DIAL), which is a novel approach to robust Active Learning (AL) in the presence of spurious bias. The proposed approach builds on previous work in passive ML setting, which, during training, assigns higher value  to underrepresented samples. DIAL, which is inspired by recent work on simplicity bias, leverages the disparity in training dynamics between over-represented and under-represented samples, thus being able to select samples that exhibit “slow” training dynamics.

**Audience:**

Yes

**Claims And Evidence:**

Yes

**Requested Changes:**

1. Most of the ABSTRACT is background, with barely 2 sentences covering DIAL (plus one sentence on experiments). Please cut in half the content before DIAL, and add 1-2 sentences on the main intuition behind DIAL (eg, the last two sentences in Section 4).

2. Same about the INTRODUCTION, where you use a full page before you get to DIAL Shorten that to half a page, and add 1-2 paragraphs on the intuition behind DIAL (right now, it only appears in Section 5).

3. In Fig 7, please remove the "Legend" from the right-most plot.

**Strengths And Weaknesses:**

STRENGTHS
==========
The paper appears to present a novel approach to AL, and it has the potential to make a practical impact. The paper is reasonably well written and easy to follow. The results in the empirical evaluation are promising, but not yet conclusive (see WEAKNESSES).

WEAKNESSES
============
1. The presentation should be improved: see requested changes in the next section.

2. The empirical validation needs to be streamlined and made more crisp:
a) In FIG 7, on all three datasets, the learning curves seem to flatten very early, way before reaching SOTA (such as [Sagawa 2020]). As the purpose of AL is to reach SOTA with fewer examples than needed by passive learning, please extent the X axes until DIAL reaches SOTA
b) in Table 1, please add the required columns to show ALC for the model built on all the training data. Without it, we cannot compare DIAL with SOTA

---

> ### Author Response · Authors · 2024-10-16
> **Response to Reviewer bAF9**
>
> We thank the reviewer for the constructive feedback. Below are our responses to the reviewer's comments.
>
> > **As the purpose of AL is to reach SOTA with fewer examples than needed by passive learning, please extent the X axes until DIAL reaches SOTA.**
>
> *short answer: SOTA performance is not comparable with AL performance in Fig. 8. Instead, We provide a comparison with the SOTA performance in Table B.2.*
>
> The purpose of AL is to reach the passive performance with fewer examples. Passive performance refers to the performance when trained on the entire dataset using ERM. In our context, the passive performance will be suboptimal due to spurious correlations. Therefore, the goal of AL is to acquire underrepresented (or minority) samples to improve the subgroup robustness of the model. In our experiments, at every AL step, the model in trained via ERM on the labelled pool, which is a standard approach in AL literature. However, the SOTA algorithms in the passive setting (e.g., [1, 2]) executes a more complex training procedure than ERM to address the bias. Therefore, we believe that it is not appropriate to compare the AL performance with that of SOTA in Fig. 8. Instead we provide a comprehensive comparison in Table B.2. in the revised manuscript.
>
> > **Summary of revision**
>
> Here is the list of changes made as per the reviewer’s request:
>
> 1. Abstract is revised to include more information about DIAL.
> 2. The content of the introduction is shortened.
> 3. Legends in the figures are adjusted to make them less obstructive to the plot.
>
> Please refer the summary of changes above for the overall changes made in the revised manuscript.
>
>
> **References**
>
> [1] E. Z. Liu, et al. Just Train Twice: Improving Group Robustness without Training Group Information. ICML 2021.
>
> [2] S. Sagawa, et al. Distributionally Robust Neural Networks for Group Shifts: On the Importance of Regularization for Worst-Case Generalization. ICLR 2020.

---

### Review · Reviewer_oT5d · 2024-10-03

**Summary Of Contributions:**

This paper investigates spurious correlations in the context of deep active learning for classification tasks. The authors analyze this concept and provide insights on leveraging the learning dynamics of deep neural networks to identify underrepresented samples. Based on this analysis, they propose a novel selection strategy specifically designed to target underrepresented or bias-conflicting samples, improving learning in such scenarios. Additionally, they present experiments demonstrating the effectiveness of their strategy in standard deep active learning settings.

**Audience:**

Yes

**Claims And Evidence:**

Yes

**Requested Changes:**

See Weaknesses above.

**Strengths And Weaknesses:**

## Strengths
- The paper is mostly well-written, with only a few minor remarks mentioned in the Weaknesses section.
- The illustrative examples effectively clarify the problem they aim to solve, making the argumentation easy to follow.
- I particularly appreciate the analysis of spurious correlations, which offers valuable insights into detecting bias-conflicting samples. Building on this, the proposed selection strategy addresses these issues. The progression from motivation to strategy is clearly and logically presented.
- The figures and the representation of the paper  really good.

## Weaknesses
- I believe the Abstract and Introduction can be improved. In the abstract, the authors assume familiarity with spurious correlations, which I did not have. A brief explanation, as provided in the introduction, would be helpful:
  > Spurious correlations refer to scenarios where certain task-irrelevant attributes in the training set are highly correlated with the target labels.

  Additionally, the explanation of the DIAL approach and the contributions could be more precise. While the authors discuss training mechanics, they could elaborate on how they capture learning dynamics, specifically through snapshots. Similarly, the contribution list could be more detailed in outlining the analysis and principles underlying DIAL.
- I believe the related work section needs revision. While the authors attempt to explain each concept, it's unclear at this stage which are crucial for the proposed strategy. For example, the neural tangent kernel framework is explained, but there's no clear indication of its relevance. Removing this section would not be an issue, as the framework can easily be discussed in later sections without introducing it in related work. *Additionally, the positioning within the literature can be improved.* For instance, the authors state:
> In contrast, our proposed method employs the standard training procedure (i.e., a typically small constant learning rate) to capture the dynamics of the learning.

  However, it's unclear why this approach is preferable to using a learning rate scheduler. In fact, this could be a limitation, as learning rate schedulers are crucial for achieving high performance.
- While the experiments on AL using subgroup performance are solid, I believe the general AL experiments need improvement. Recent literature in AL [1,2] highlights the significance of pretrained foundation models, showing that these models are critical for AL, with factors such as representativeness and informativeness having a much larger impact than training models from scratch. Moreover, fine-tuning a pretrained model is more realistic than training from scratch on vision datasets. I suggest replacing or adding experiments that incorporate these models. Fine-tuning just the last linear layer would make the experiments fast and more realistic.
- Lastly, I recommend that the authors highlight the limitations of their approach. While DIAL performs well in their experiments, it requires a constant learning rate, which is not optimal in deep learning, and its computational complexity is higher than other strategies due to the need for inference through an ensemble. Including a limitations paragraph of section that addresses these issues is important in my opinion.

Some minor remarks:
- The authors frequently cite sources without parentheses when they should be enclosed. For example, in the Related Work section, "Gal & Ghahramani (2016)" or "in Jung et al. (2022)" should use parentheses. Similarly, in the Background section, "the predictive error Sagawa et al. (2020); Liu et al. (2021); Duchi & Namkoong (2021)" is missing parentheses. I recommend thoroughly reviewing all references and updating them accordingly.
- In related work there is a missing period after "Active learning for fairness".
- I am not sure why the authors mention out-of-distribution problems in the introduction. The wording should be optimized, as it is only briefly discussed in the Introduction and Abstract, without being addressed further in the paper.

[1] Guy Hacohen, Avihu Dekel, and Daphna Weinshall. Active learning on a budget: Opposite strategies suit high and low budgets. In International Conference on Machine Learning, pp. 8175–8195, 2022.

[2] Sanket Rajan Gupte, Josiah Aklilu, Jeffrey J. Nirschl, and Serena Yeung-Levy. Revisiting Active Learning in the Era of Vision Foundation Models. Transactions on Machine Learning Research, 2024.

---

> ### Author Response · Authors · 2024-10-16
> **Response to Reviewer oT5d**
>
> We thank the reviewer for the constructive feedback. Below are our responses to the reviewer's comments.
>
> > **However, it’s unclear why this approach is preferable to using a learning rate scheduler. In fact, this could be a limitation, as learning rate schedulers are crucial for achieving high performance.**
>
> *short answer: we do not use a cyclic LR scheduler for the acquisition phase in order to capture the dynamics of learning. The statement for the constant learning rate is a mistake.*
>
> We appreciate the reviewer’s concern. The reason we do not use the cyclic LR scheduler is that we aim to capture the dynamics of the learning process for the acquisition purposes. For example, [5] uses the cyclic learning rate scheduler to generate diverse checkpoints to replace the ensemble. However, our aim is not to achieve diversity in the ensemble, but to allow the checkpoints to represent snapshots of the learning dynamics.
>
> Besides that,we would like to correct an error in our manuscript regarding the fixed learning rate. We did not intend to imply that the learning rate remains constant throughout training, as is the case with optimizers like SGD. Rather, we mean to convey that we do not utilise a cyclic learning rate scheduler. In fact, in our experiments, we used the ADAM optimiser, which uses an adaptive learning rate, for the Corrupted CIFAR-10 dataset. We apologise for the confusion and have clarified this in the revised manuscript. Finally, we also want to highlight that using the SGD optimiser is a common practice for subpopulation shift datasets such as Waterbirds and CelebA datasets [2, 1, 3].
>
> > **Foundation models**
>
> We appreciate the reviewer’s suggestion. The focus of this work is to introduce a AL method aimed at acquiring samples that enhance the subgroup robustness of the model. We position our work as a counterpart to subgroup robustness methods from the passive learning framework. To that end, we compared the SOTA baselines [3], which are typically built on pretrained models, such as ImageNet pretrained ResNets-50. While using foundation models could potentially enhance performance further, exploring this is beyond the scope of in this work. Moreover, the previous study [4] suggests that foundation models may not offer substantial advantages in specific subpopulation shifts, such as spurious correlations. We have acknowledged this suggestion as a direction for future work in the revised manuscript.
>
>
> > **Summary of revision**
>
> Here is the list of changes made as per the reviewer’s request:
>
> 1. Minor issues such as typos, citations, formatting, terminology etc. are fixed.
> 2. Abstract, related work and introduction are shortened.
> 3. The discussion of limitations is added in the conclusion (Section 8).
>
> Please refer the summary of changes above for the overall changes made in the revised manuscript.
>
> **References**
>
> [1] E. Z. Liu, et al. Just Train Twice: Improving Group Robustness without Training Group Information. ICML 2021.
>
> [2] S. Sagawa, et al. Distributionally Robust Neural Networks for Group Shifts: On the Importance of Regularization for Worst-Case Generalization. ICLR 2020.
>
> [3] Y. Yang, et al. Change is Hard: A Closer Look at Subpopulation Shift. ICML 2023.
>
> [4] M. Zhang and C. Ré. Contrastive Adapters for Foundation Model Group Robustness. NeurIPS 2022.
>
> [5] S. Jung,et al. A Simple Yet Powerful Deep Active Learning With Snapshots Ensembles. ICLR 2022.

---

### Review · Reviewer_kRh4 · 2024-10-03

**Summary Of Contributions:**

**Problem Setting:** Spurious attributes within the training set can lead to biased deep neural networks (DNNs) with low generalization performances. As a result, several works address this issue in a passive learning setting [1, 2], e.g., by reweighting underrepresented samples during the training. In contrast, the authors of this submission showcase that instead of reweighting samples, *active learning* (AL) can effectively reduce spurious attributes' negative impact already during the data collection.

**Proposed Solution:** The author's central idea is to prioritize actively selecting underrepresented samples from the unlabeled pool for annotation. These underrepresented samples are assumed to be *bias-conflicting* (BC) samples, whose incorporation into the labeled pool as a training set reduces the negative impact of *bias-aligned* (BA) samples on the DNN's generalization performance. Based on observations from the literature [3], the authors argue that sample-wise training loss curves can serve as indicators for BC samples. In other words, DNNs tend to learn BC samples more slowly than BA samples.
For this reason, the proposed strategy *domain-invariant active learning* (DIAL) selects samples for which members of a snapshot ensemble disagree most. An empirical evaluation finds that DIAL's performance is superior or competitive compared to other (partially state-of-the-art) AL strategies on datasets with and without spurious bias. A final ablation study analyzes the effects of the number of snapshot ensemble members, query batch size, and functions to measure disagreements as central components of DIAL.

**References:**
- [1] Liu, Evan Z., et al. Just train twice: Improving group robustness without training group information. ICML 2021.
- [2] Nam, Junhyun, et al. Learning from failure: De-biasing classifier from biased classifier. NeurIPS 2020.
- [3] Shah, Harshay, et al. The pitfalls of simplicity bias in neural networks. Neurips 2020.

**Audience:**

Yes

**Broader Impact Concerns:**

I do not see any direct ethical concerns associated with the submission. However, spurious biases themselves can have severe ethical implications in critical application domains. Adding a brief statement as part of a **Broader Impact** section regarding this issue and mentioning that the proposed AL strategy can be helpful in this context but does not give any theoretical guarantees would embed the work in a broader context.

**Claims And Evidence:**

No

**Requested Changes:**

The previously outlined **Strengths** indicate that the submission presents a highly relevant topic with an interesting solution. Yet, claims, such as computational efficiency, and a missing discussion of limitations reveal that further evidence to strengthen this work is required. Beyond potential changes already proposed in the Section **Weaknesses**, I, therefore, suggest that the authors make the following adjustments to their paper:

- Make another iteration to check for
  - typos, e.g., "Empirical results demonstrate~s~" in the Abstract,
  - missing brackets around references,
  - a consistent list of references, e.g., "Proceedings of The 33rd International Conference on Machine Learning" vs. "International Conference on Machine Learning",
  - and inconsistent terms, e.g., AL (acquisition) method vs. AL strategy.
- Differences in the learning curves, e.g., in Fig. 9 (b), are barely visible. Instead of showing the absolute accuracy values, the authors could plot the accuracy differences between AL strategies and random sampling to make the plots more clear.
- Although there are several examples illustrating the negative effects of spurious bias, e.g., car classification, two-dimensional dataset, and colored MNIST, the switching between these examples makes it sometimes difficult to follow. In my opinion, introducing the MNIST example already in the introduction with visualizations would ease the problem understanding from the beginning.
- I would expect that the labeled pool $\mathcal{D}_L$ contains tuples of samples and labels. In this case, the set operation $\mathcal{D}_L \cup \mathcal{Q}$ is not defined if the unlabeled pool $\mathcal{D}_U$ contains only samples (no labels), of which the query set is a subset $\mathcal{Q} \subset \mathcal{D}_U$.
- The learning curve figures, e.g. Fig. 12, would benefit from clear legends for the y-axes.

**Remark:** In case, I overlooked or misunderstood anything of the work, I'm happy to be corrected and adjust my review accordingly. Looking forward to a constructive discussion.

**Strengths And Weaknesses:**

**Strengths:**
- The authors tackle the important problem of spurious bias in training data, which seems to be underexplored in an AL setting.
- The idea of the proposed strategy DIAL is straightforward and motivated by well-established observations from the literature and own examples/studies in combination with nice illustrations.
- The performance of DIAL is empirically evaluated and compared to well-established AL strategies on common benchmark datasets, with and without spurious bias. The obtained results seem to be promising.
- An ablation study explains the design choices regarding the different components of DIAL.
- Not only accuracy results are reported, but there are also detailed analyses of the AL strategies' sampling behaviors with respect to the different subgroups in datasets with spurious biases.

**Weaknesses:**
- The optimizer seems to play an important role in the generation of snapshot ensembles and, thus, could impact the performance of DIAL. The authors partly investigate this impact by comparing DIAL using a fixed learning rate to Jan et al. [4] who employ cosine annealing as a learning rate scheduler. Although the authors correctly state that fixed learning rates are a standard training procedure (cf. end of first paragraph in Section 2), I would say that learning rate schedulers are also pretty standard in deep learning research [5, 6].  Accordingly, requiring a fixed learning rate to reliably identify BC samples can be seen as a restriction of DIAL. Moreover, many optimizers beyond vanilla *stochastic gradient descent* (SGD), e.g., Adam, adjust the learning rate for each weight of a DNN dynamically. The hyperparameter table in Appendix A.2 indicates that Adam has also been used for the training on Corrupted CIFAR-10. Therefore, it is unclear how this optimizer matches the fixed learning rate requirement in Section 5.3.
- The computational complexity is indeed an important aspect in AL, where all kinds of costs are to be minimized. Despite the statement that DIAL needs no "resource-intensive computations" (cf. Abstract), there is no further empirical or theoretical evidence in favor of this aspect. For example, I would have expected results on the selection times for the AL strategies or even a comparison using the $\mathcal{O}$-notation. Regarding DIAL, I think that the feed-forward passes for large network architectures and many snapshot ensembles can be computationally expensive.
- The ablation study delivers many insights to better understand the behavior of DIAL. Yet, it is unclear to me, which and how the authors finally selected the number of snapshot members $M$ and the distance function $\phi$ to compare DIAL to its competitors.
- There is no extensive discussion of potential limitations of the author's work. For example, despite the promising results of DIAL, this strategy is rather empirically motivated lacking theoretical guarantees and requires a suitable choice for the number of snapshot ensemble members. There is no need to resolve such limitations as part of this work, but it is necessary to transparently discuss them.
- I appreciate that the authors detail the related software packages and repositories used in their empirical evaluation. However, I could not find any code to review (e.g., as attached supplementary material or via Anonymous GitHub) or indication that code will be made publicly in case of acceptance.
- Averaging accuracies across datasets (as done in Table 1) is subjective to outliers [8]. Computing a ranking for each dataset and averaging these ranks could be a more robust alternative.

**Question:** The formal definition of spurious correlation in Eqs. (2) and (3) is unclear to me because the joint distribution $P(X, Y, S)$ of sample attributes $X$, spurious attributes $S$, and true class labels $Y$ is claimed to be proportional to an expression, which is independent of the sample attributes $X$. Reading the referred work of Yang et al. [7] helped to better understand the formalization of spurious correlation (cf. Eq. (1) and Table 1 in [7]). Can the authors provide an explanation to resolve this unclarity from my side?

**References:**
- [4] Jung, Seohyeon, Sanghyun Kim, and Juho Lee. A simple yet powerful deep active learning with snapshots ensembles. ICLR 2023.
- [5] Gotmare, Akhilesh, et al. A Closer Look at Deep Learning Heuristics: Learning rate restarts, Warmup and Distillation. ICLR 2019.
- [6] He, Kaiming, et al. Deep residual learning for image recognition. CVPR 2016.
- [7] Yang, Yuzhe, et al. Change is Hard: A Closer Look at Subpopulation Shift. ICML 2023.
- [8] Demšar, Janez. Statistical comparisons of classifiers over multiple data sets. JMLR 2006.

---

> ### Author Response · Authors · 2024-10-16
> **Response to Reviewer kRh4**
>
> We thank the reviewer for the constructive feedback. Below are our responses to the reviewer's comments.
>
>
> > **Requiring a fixed learning rate to reliably identify BC samples can be seen as a restriction of DIAL.**
>
> *short answer: we do not use a cyclic LR scheduler for the acquisition phase in order to capture the dynamics of learning. The statement for the constant learning rate is a mistake.*
>
> We appreciate the reviewer’s concern. The reason we do not use the cyclic LR scheduler is that we aim to capture the dynamics of the learning process for the acquisition purposes. For example, [4] uses the cyclic learning rate scheduler to generate diverse checkpoints to replace the ensemble. However, our aim is not to achieve diversity in the ensemble, but to allow the checkpoints to represent snapshots of the learning dynamics.
>
> Besides that,we would like to correct an error in our manuscript regarding the fixed learning rate. We did not intend to imply that the learning rate remains constant throughout training, as is the case with optimizers like SGD. Rather, we mean to convey that we do not utilise a cyclic learning rate scheduler. In fact, in our experiments, we used the ADAM optimiser, which uses an adaptive learning rate, for the Corrupted CIFAR-10 dataset. We apologise for the confusion and have clarified this in the revised manuscript. Finally, we also want to highlight that using the SGD optimiser is a common practice for subpopulation shift datasets such as Waterbirds and CelebA datasets [2, 1, 3].
>
> > **Computational complexity**
>
> *short answer: DIAL only requires extra forward passes during the acquisition phase. The computation for the acquisition score is almost negligible in most cases. A discussion has been added in the revised manuscript in Section 8.*
>
> We appreciate the reviewer’s concern. Unfortunately, we do not have a rigorous theoretical analysis for the computational complexity of DIAL. However, we would like to provide some brief analysis for better understanding. Assuming the size of unlabelled pool $|\mathcal D_U|= n$, DIAL’s computational complexity is similar to those ensemble-based AL methods, as it requires extra forward passes for each ensemble member during the acquisition phase, thus the complexity for the forward passes will be scale by a factor of $m$: $\mathcal O(mn)$, where $m$ is the number of the checkpoints in $\mathcal M$.
>
> In the acquisition phase, for each sample, DIAL requires to compute pairwise similarities (e.g., Euclidean distance) between the predictions of checkpoints: $\phi(f_{\theta}(x),f_{\theta'} (x))$, for all $(\theta,\theta') \in \mathcal M$. The complexity for this computation is $\mathcal O(m^2n)$. Therefore, the overall complexity will be $\mathcal O(mn+ m^2n)$. However, DIAL only processes the DNN’s output $f_{\theta}(x)$ when computing the acquisition scores, which is typically a vector of size $|\mathcal Y|$ (number of classes), and this is usually quite small (e.g., 10 for CIFAR-10). Hence, the execution time for computing acquisition scores is minimal if using vectorised operations. Therefore, when the dimension of $f_\theta(x)$ is small, the complexity of DIAL is primarily dominated by the forward passes, which is approximately $\mathcal O(mn)$.
>
> As demonstrated in our ablation study, mdoes not need to be extremely large to achieve good performance. Besides that, as DIAL’s complexity only scales linearly with $n$, so we believe DIAL is still considered relatively efficient and fit well for the deep learning framework than other methods that require costly computations which can scale up to $\mathcal O(n^3)$ [5].

---

> > ### Author Response · Authors · 2024-10-16
> > **Response to Reviewer kRh4 (Cont'd)**
> >
> > > **It is unclear to me, which and how the authors finally selected the number of snapshot members and the distance function to compare DIAL to its competitors.**
> >
> > As of now, we do not have a procedure for selecting the number of snapshot members prior to the AL process. However, we have found that using 10-20 checkpoints is effective for a total of 1,000 training steps, while 30-50 checkpoints work well for training steps up to 5,000. A statement regarding this limitation has been added to the revised manuscript in Section 8. Regarding the choice of the distance function, we found that the $L_2$ norm consistently yields more accurate detection of predictive discrepancies across all datasets compared to other probabilistic divergence metrics. We have included the information about the number of checkpoints and the distance function used in our experiments in the revised manuscript in Section 7.
> >
> > > **The formal definition of spurious correlation in Eqs. (2) and (3) is unclear.**
> >
> > From the data generation perspective, the joint distribution $P(X,Y,S)$ can be expressed as $P(X | Y,S)P(Y | S)P(S)$. In our problem, we assume that the covariate $P(X | Y,S)$ is invariant across training and test distributions, i.e., $P_{tr}(X | Y,S) = P_{te}(X | Y,S)$, hence it is omitted in the expressions, we added a footnote to clarify this in the revised manuscript. In training set, because of the statistical dependency between $Y$ and $S$, $P(Y | S)  \neq P(Y)$. For example, likelihood of observing a specific class in the dataset changes when conditioned on the spurious label, which is the case of spurious correlation. While in the test set where spurious correlation is absent, $P(Y | S) = P(Y)$, implying independence between $Y$ and $S$. We hope this clarifies the definition of spurious correlation.
> >
> > > **Summary of revision**
> >
> > Here is the list of changes made as per the reviewer’s request:
> >
> > 1. Minor issues such as typos, citations, plotting, bibliography, terminology etc. are fixed.
> > 2. y-axis in Fig. 10b is replaced with the relative improvement for better visibility.
> > 3. The MNIST example (with visualizations) is introduced in the introduction section.
> > 4. The set annotation for $\mathcal Q$ is fixed.
> > 5. The discussion of limitation is added in Section 8.
> > 6. A link to the anonymous repository is added in Appendix A.2.
> > 7. The average accuracy across datasets in Table 1 is replaced with the averaged ranking.
> > 8. A broader impact statement is added.
> >
> > Please refer the summary of changes above for the overall changes made in the revised manuscript.
> >
> >
> > **References**
> >
> > [1] E. Z. Liu, et al. Just Train Twice: Improving Group Robustness without Training Group Information. ICML 2021.
> >
> > [2] S. Sagawa, et al. Distributionally Robust Neural Networks for Group Shifts: On the Importance of Regularization for Worst-Case Generalization. ICLR 2020.
> >
> > [3] Y. Yang, et al. Change is Hard: A Closer Look at Subpopulation Shift. ICML 2023.
> >
> > [4] S. Jung,et al. A Simple Yet Powerful Deep Active Learning With Snapshots Ensembles. ICLR 2022.
> >
> > [5] J. Ash, et al. Gone Fishing: Neural Active Learning with Fisher Embeddings. NeurIPS 2021.

---

### Review · Reviewer_zGQQ · 2024-10-03

**Summary Of Contributions:**

Models trained with standard ERM techniques pick up on spurious correlations present in the training data. Techniques that attempt to mitigate this tend to rely on labels, and are therefore not applicable to the active learning (AL) setting where we care about selecting a subset from an unlabeled pool. This paper claims that existing AL acquisition functions do not prevent models from learning spurious correlations, but the newly proposed approach DIAL has the ability to do so. DIAL operates based on the observation that samples with spurious features are learnt slower than samples without them, and uses a heuristic to identify the slowly learnt samples. Experiments show that DIAL can outperform some existing AL strategies in both standard settings and settings were spurious correlations are present.

**Audience:**

Yes

**Claims And Evidence:**

No

**Requested Changes:**

## Requested Changes
**Synthetic Experiment is unclear**
- It is unclear how Fig. 1 highlights a flaw in the margin score. Given that the classifier is already achieving ~100% accuracy, it is unclear what a good batch of samples would even look like here. In practice, we would have already ended the active learning loop before even getting to this stage.
- In this case, it seems like adding more samples in the minor modes (BC samples) would not really change the decision boundary so there is not much utility in selecting these.
- Even if this were a flaw, this can trivially be resolved with the use of acquisition functions that incorporate diversity which has not been discussed in this subsection.
- The BA/BC terms are confusing, I would encourage the authors to use some more intuitive terminology in this section.

**Using fixed learning rates its highly unrealistic**
- This is mentioned in the description of the approach. Is this a hard requirement? If so, this significantly limits the applicability of this approach. It is most likely possible to see higher performance gains with a better learning rate schedule and an inferior acquisition function.
- Experiments should always use the most optimal learning rate schedule.

**Full gradients can be impractical**
- Using/storing full gradients + computing dot products can be expensive in the context of massive models with billions of parameters for a large scale unlabeled dataset. Is it possible to use approximations (use gradients of only the last layer as in [3]) or perhaps only using softmax predictions of the model?
- Some discussion and experiments with gradient approximations are important for this work be extended to the real world.

**Missing comparison in Fig. 6**
- It is well-known that uncertainty based methods will select imbalanced batches, but again, acquisition functions that use diversity may not have this issue.
- It is unclear why the balance of the batch is important, seems like the distribution of the whole labeled pool is what we care about since that is what will affect downstream performance.

**Experiments need far more baseline comparisons**:
- The experiments do not compare against recent approaches in active learning; the most recent baseline is from 2020 (Badge [Ash et al.'20]). Given that most of the motivating examples and intuition only applies to uncertainty based AL, more baseline comparisons are needed. See BAIT [3], Cluster-Margin [4], FL [5].
- Some comparison against other work that leverages training dynamics is needed as well, given that this is not a new idea in the field of AL.

**Discussion of Loss-Estimation Based AL is needed**
- In active learning, while it is not possible to get the true loss since labels are not present, many works such as [2] learn to estimate loss during training. Is it possible for such methods to be combined with existing methods for eliminating spurious correlations that require labels?
- Some discussion of this is needed given the popularity of this approach, and ideally an empirical comparison is needed.

**Plotting Issues**:
- In Figures 2, 7, 8, 9, the legend is obscuring a large part of the plot which hinders readability. Please improve legend placement in the next version of the paper.
- Instead AL step on the x-axis, please report the size of the labeled set.

**Minor Terminology Issue**:
-  The paper refers to selecting samples with low confidence as "certainty" sampling though this is typically referred to as **confidence sampling**. I would recommend changing this in the plots and baseline descriptions

**Typos**:
- distinctions, though, are blur. -> distinctions, though, are **blurry**.
- Tracking tranining trajectory -> Tracking **training** trajectory

**References**
- [1] Coresets for Data-efficient Training of Machine Learning Models, ICML 2019
- [2] Learning Loss for Active Learning, CVPR 2019
- [3] Gone Fishing: Neural Active Learning with Fisher Embeddings, NeurIPS 2021
- [4] Batch Active Learning at Scale, NeurIPS 2021
- [5] An Experimental Design Framework for Label-Efficient Supervised Finetuning of Large Language Models, ACL Findings 2024

**Strengths And Weaknesses:**

## Strengths:
This work studies a frequently overlooked aspect of active learning. While most work focuses purely on accuracy, this work also considers ERM's tendency to learn spurious correlations.


## Weaknesses:
While I can imagine that this problem persists with uncertainty-based active learning, state of the art active learning techniques incorporate both diversity and uncertainty and will likely not suffer from this weakness. However, the motivating examples solely focus on uncertainty and the experimental results only assess outdated diversity-based AL techniques. Finally, some of the design choices such as fixed learning rate and using the full gradient need to be analyzed with more experiments and ablations. See requested changes for specifics.

---

> ### Author Response · Authors · 2024-10-16
> **Response to Reviewer zGQQ**
>
> We thank the reviewer for the constructive feedback. Below are our responses to the reviewer's comments.
>
>
> > **Synthetic Experiment is unclear.**
>
> - Classifiers shown in Fig. 2 (in the revised manuscript) are not optimal and do not achieve 100% accuracy (a statement for optimal classifier has been added in the revised manuscript). The classifiers are relying on the spurious feature (i.e., colours) to make the predictions. Hence, the off-diagonal subgroups (bias-conflicting (BC)) are always misclassified. The good batch of samples will be the ones from the BC subgroups, as they are the ones misrepresenting in the labelled pool. We have added a paragraph to clarify this in the revised manuscript.
> - We added a classifier in Fig. 2 to show the hypothetical scenario where all the BC samples are acquired. From completely misclassifying the BC samples, the classifier becomes able to correctly classify a substantially large amount of BC samples. This is a sign of the improvement of subgroup robustness, which further validates our argument of the necessity of favouring the BC samples.
> - We added the diversity-based approach in Fig. 2 and a paragraph in Section 4 explaining why diversity is not sufficient for the problem. We agree with the reviewer that the terminology requires improvement to help readers to better understand the problem. Hence, in the revised manuscript, we detailed the assignment of the BA/BC subgroups in the synthetic experiment and used more intuitive terms minority and majority
> to refer to BC and BA subgroups, respectively.
>
> > **Using fixed learning rates is highly unrealistic.**
>
> *short answer: we do not use a cyclic LR scheduler for the acquisition phase in order to capture the dynamics of learning. The statement for the constant learning rate is a mistake.*
>
> We appreciate the reviewer’s concern. The reason we do not use the cyclic LR scheduler is that we aim to capture the dynamics of the learning process for the acquisition purposes. For example, [4] uses the cyclic learning rate scheduler to generate diverse checkpoints to replace the ensemble. However, our aim is not to achieve diversity in the ensemble, but to allow the checkpoints to represent snapshots of the learning dynamics.
>
> Besides that,we would like to correct an error in our manuscript regarding the fixed learning rate. We did not intend to imply that the learning rate remains constant throughout training, as is the case with optimizers like SGD. Rather, we mean to convey that we do not utilise a cyclic learning rate scheduler. In fact, in our experiments, we used the ADAM optimiser, which uses an adaptive learning rate, for the Corrupted CIFAR-10 dataset. We apologise for the confusion and have clarified this in the revised manuscript. Finally, we also want to highlight that using the SGD optimiser is a common practice for subpopulation shift datasets such as Waterbirds and CelebA datasets [1, 2, 3].
>
> > **Full gradients can be impractical.**
>
> *short answer: DIAL does not require any gradient computation.*
>
> We believe that the reviewer has misunderstood Eq. (4). The notation $f_\theta(x)$ refers to the output of the model $$f with parameters $\theta$ for the input $x$. The measure of disagreement is computed based on the $f_\theta$ from the training checkpoints $\theta \in \mathcal M$. Hence, the criteria (e.g., dot product, etc) is computed based on the model’s output, not the gradients, which is usually in the size of the number of classes. Please refer to our response to Reviewer kRh4 for the discussion on the computational complexity of DIAL.
>
> > **Missing comparison in Fig. 6 and it is unclear why the balance of the batch is important.**
>
> The subgroup robustness performance is measured by the worst-group accuracy [2], i.e., the accuracy of the worst-performing subgroup. Typically, there exists multiple underrepresented subgroups in the dataset, using methods that yield imbalanced batches, only certain underrepresented subgroups will be favoured, while the others will be neglected, as a result, the worst-group accuracy will still likely maintain low, and no improvement in robustness is achieved. Hence, the balance of the batch is a crucial factor for the subgroup robustness. The purpose of Fig. 7 (in the revised manuscript) is to demonstrate the imbalanced of the informativeness using uncertainty-based methods (e.g., Margin). We have added the CoreSet (diversity-based strategy) in Fig. 2 and discussion on the impact of balance of the batch in Section 4. The title for the paragraph for Fig. 7 has been revised for better clarity.

---

> > ### Author Response · Authors · 2024-10-16
> > **Response to Reviewer zGQQ (Cont'd)**
> >
> > > **Experiments need far more baseline comparisons.**
> >
> > We appreciate the reviewer’s suggestion. Two baselines (BAIT [5] and Cluster-Margin [6]) have been added in the revised manuscript. Unfortunately, we are not able to include the result of BAIT on Corrupted-CIFAR10 and two other standard AL datasets, due to computational constraints, as it took us five hours to run only three AL iterations; manipulating the kernel matrix is resource-intensive for large datasets. Based on the updated results, DIAL outperforms these two baselines on the datasets with spurious correlations.
> >
> > > **Discussion of Loss-Estimation Based AL is needed**
> >
> > - If we understand the reviewer’s first question correctly,it asks whether the loss-estimation-based AL approach can be adapted to address subpopulation shift problem in the passive setting where the spurious labels are not available. In fact, there are two SOTA (passive learning) baselines [1, 11] that are based on the similar
> > intuition, where minority samples are identified by the loss or classification error(without the spurious labels) and then upsampled to improve the model’s robustness.
> >
> > - If the reviewer is asking about the potential of using the loss-estimation-based AL approach [7] to address the subpopulation shift problem in the AL setting, we believe that it may encounter the same challenges as ERM when spurious correlations are present. Specifically, ERM struggles to provide accurate predictions for minority subgroups due to their underrepresentation in the training data. A similar problem could arise when learning and estimating the loss where the model might fail to accurately estimate the loss for the minority subgroups due to their underrepresentation.
> >
> > If the reviewer finds our response satisfactory, we will include this discussion in the revised manuscript.
> >
> > > **Plotting issues: x-axis.**
> >
> > The reason we used the AL step number as the x-axis is because the datasets are in different scales. So the learning curve figures will look consistent across different datasets. We also noticed that some prior works also use the AL step number (or similar terminology) as the x-axis [8, 9, 10].
> >
> > **Summary of revision**
> >
> > Here is the list of changes made as per the reviewer’s request:
> >
> > 1. Minor issues such as typos, plotting, terminology etc. are fixed.
> > 2. The synthetic experiment in Fig. 2 is revised for better clarity.
> > 3. Discussion on the limitation of diversity-based approach is added in Section 4.
> > 4. Two baselines (BAIT and Cluster-Margin) are added for comparison.
> > 5. The terminology is improved in Section 4.
> >
> > Please refer the summary of changes above for the overall changes made in the revised manuscript.
> >
> >
> > **References**
> >
> > [1] E. Z. Liu, et al. Just Train Twice: Improving Group Robustness without Training Group Information. ICML 2021.
> >
> > [2] S. Sagawa, et al. Distributionally Robust Neural Networks for Group Shifts: On the Importance of Regularization for Worst-Case Generalization. ICLR 2020.
> >
> > [3] Y. Yang, et al. Change is Hard: A Closer Look at Subpopulation Shift. ICML 2023.
> >
> > [4] S. Jung,et al. A Simple Yet Powerful Deep Active Learning With Snapshots Ensembles. ICLR 2022.
> >
> > [5] J. Ash, et al. Gone Fishing: Neural Active Learning with Fisher Embeddings. NeurIPS 2021.
> >
> > [6] G. Citovsky, et al. Batch Active Learning at Scale. NeurIPS 2021
> >
> > [7] D.Yoo and I.S.Kweon. Learning Loss for Active Learning. CVPR 2022
> >
> > [8] C. Riis, et al. Bayesian Active Learning with Fully Bayesian Gaussian Processes, NeurIPS 2022.
> >
> > [9] H. Wang, et al. Deep Active Learning by Leveraging Training Dynamics. NeurIPS 2022
> >
> > [10] M. A. Mohamadi, et al. Making Look-Ahead Active Learning Strategies Feasible with Neural Tangent Kernels. NeurIPS, 2022.
> >
> > [11] J. Nam, et al. Learning from Failure: De-biasing Classifier from Biased Classifier. NeurIPS, 2020.

---

### Decision · Action_Editor_cCSr · 2025-01-07

**Recommendation:** Reject

**Comment:**

The manuscript was reviewed by four reviewers [zGQQ,kRh4,oT5d,bAF9], and there was discussion between reviewers and authors, and corresponding updates to the manuscript. However, although the reviewers appreciated the updates and felt they improved the manuscript, all four reviews of this manuscript nevertheless ultimately leaned towards rejection.

In the original reviews, several aspects of the work were appreciated:
+ Studying an overlooked aspect of active learning was appreciated [zGQQ]
+ The problem was considered important [kRh4] and the paper was considered to have potential for practical impact [bAF9]
+ The idea of the strategy was considered straightforward [kRh4] and motivated [kRh4,oT5d]
+ The approach was considered novel [bAF9]
+ The results were considered promising [kRh4,bAF9]
+ The illustrative examples and analysis of spurious correlations were appreciated [oT5d]
+ The ablation study and analyses of active learning sampling behaviors were appreciated [kRh4]
+ The paper was considered well writen [oT5d,bAF9] and well presented [oT5d]

However, several concerns were brought up:
- One reviewer felt state of the art active learning techniques would not suffer from the motivating problem [zGQQ]
- The match between the Adam optimizer and the fixed learning rate requirement was considered unclear [kRh4]
- The synthetic experiment was considered unclear [zGQQ]
- Use of fixed learning rates was considered unrealistic [zGQQ]
- Extending learning curves until DIAL reached state of the art, and ALC values for the model built on all training data, were desired [bAF9]
- Additional analyses, experiments and ablation studies for design choices were desired [zGQQ]
- Empirical results on selection times or comparisons of computation time O-notations were desired [kRh4]
- Computational load of using full gradients was a concern [zGQQ]
- Comparison to acquisition functions using diversity was desired [zGQQ]
- More baselines representing recent approaches were desired [zGQQ,oT5d] including fine-tuned pretrained models [oT5d]
- Information on selection of the number of snapshot members in the ablation study was desired [kRh4]
- Discussion of loss estimation approaches during the active learning was desired [zGQQ]
- Outlier impact on averaged accuracies was a concern [kRh4]
- Early explanation of spurious correlations was desired [oT5d]
- More precise explanation of DIAL [oT5d,bAF9] and the contributions was desired [oT5d]
- Clarification of relevances of the related work and improved positioning of the current work among the existing work was desired [oT5d]
- Additional clarity improvements in result presentation and some clarifications were desired [kRh4]
- Additional discussion of limitations was desired [kRh4,oT5d]
- Code for review was desired [kRh4]

The authors provided responses to the concerns. However, despite the discussion and updates, several concerns still remained:

- The motivation of the work was still considered unconvincing [zGQQ]
- Reviewers remained concerned that the lack of learning rate schedulers was a limitation that was not sufficiently addressed [kRh4,oT5d] and can yield subpar performance [kRh4]
- The empirical evidence was considered weak [bAF9] and comparison to DRO was desired [bAF9]
- An evaluation of DIAL without a learning rate scheduler in the selection model but then training a standard neural network classifier with a learning rate scheduler was desired [kRh4]
- The synthetic experiments were still considered unclear [zGQQ]
- Use of a poor initial classifier was seen as not well motivated and use of e.g. diversity based sampling was desired [zGQQ]
- Testing of the heuristic for the |M| hyperparameter was considered unclear and potentially yielding unfair comparisons to strategies not needing hyperparameter tuning [kRh4].

Due to the strong remaining concerns, I must agree with the reviewers that although the manuscript could become a good contribution in time, the manuscript does not yet seem to by supported by clear enough evidence to be acceptable to TMLR in its current state.

**Audience:**

The topic of analyzing active learning under spurious bias and proposing an active learning algorithm robust to spurious bias was seen as an important problem and an overlooked aspect of active learning, thus it seems clear that some in the TMLR audience would be interested in the work.

**Claims And Evidence:**

Despite the revisions made by the authors, the manuscript does not yet seem to by supported by clear enough evidence to be acceptable to TMLR in its current state; several concerns regarding motivationi and empirical evidence remain, see the Comment field for details.

**Resubmission Of Major Revision:**

The authors may consider submitting a major revision at a later time.